# Immunotherapeutic Strategies for the Treatment of Glioblastoma: Current Challenges and Future Perspectives

**DOI:** 10.3390/cancers16071276

**Published:** 2024-03-25

**Authors:** Ilaria Salvato, Antonio Marchini

**Affiliations:** 1NORLUX Neuro-Oncology Laboratory, Department of Cancer Research, Luxembourg Institute of Health (LIH), L-1210 Luxembourg, Luxembourg; msallaria@gmail.com; 2Laboratory of Oncolytic Virus Immuno-Therapeutics (LOVIT), Department of Cancer Research, Luxembourg Institute of Health (LIH), L-1210 Luxembourg, Luxembourg; 3Department of Life Sciences and Medicine, Faculty of Science, Technology and Medicine (FSTM), University of Luxembourg, L-4367 Belvaux, Luxembourg; 4Laboratory of Oncolytic Virus Immuno-Therapeutics, German Cancer Research Center, 69120 Heidelberg, Germany

**Keywords:** GBM, GBM immunosuppressive tumor microenvironment, immunotherapy, immune checkpoint therapy, adoptive cell therapy, vaccination therapy, DNA/RNA vaccines, CAR-T cell therapy, oncolytic virotherapy

## Abstract

**Simple Summary:**

Glioblastoma (GBM) poses a formidable challenge as a central nervous system tumor with extremely limited responsiveness to conventional treatments. While immunotherapeutic approaches have shown success in treating other solid tumors, their effectiveness against GBM is limited. Our review systematically addresses the intrinsic features of GBM that hinder the success of both standard therapies and immunotherapies. Furthermore, we comprehensively analyze all the immune-based approaches currently undergoing clinical evaluation for GBM, both as standalone treatments and in combination with standard therapy or other immunotherapies.

**Abstract:**

Despite decades of research and the best up-to-date treatments, grade 4 Glioblastoma (GBM) remains uniformly fatal with a patient median overall survival of less than 2 years. Recent advances in immunotherapy have reignited interest in utilizing immunological approaches to fight cancer. However, current immunotherapies have so far not met the anticipated expectations, achieving modest results in their journey from bench to bedside for the treatment of GBM. Understanding the intrinsic features of GBM is of crucial importance for the development of effective antitumoral strategies to improve patient life expectancy and conditions. In this review, we provide a comprehensive overview of the distinctive characteristics of GBM that significantly influence current conventional therapies and immune-based approaches. Moreover, we present an overview of the immunotherapeutic strategies currently undergoing clinical evaluation for GBM treatment, with a specific emphasis on those advancing to phase 3 clinical studies. These encompass immune checkpoint inhibitors, adoptive T cell therapies, vaccination strategies (i.e., RNA-, DNA-, and peptide-based vaccines), and virus-based approaches. Finally, we explore novel innovative strategies and future prospects in the field of immunotherapy for GBM.

## 1. Introduction

Glioblastoma (GBM) is the most aggressive primary brain tumor, accounting for nearly 50% of all the primary central nervous system malignancies [1,2]. GBMs develop spontaneously within the brain (de novo) and typically infiltrate nearby brain tissues without spreading to distant organs [3]. Its incidence is 3.23 per 100,000 persons in the United States, with a slightly higher occurrence in males compared to females [4]. It is a fast-growing tumor occurring in patients with an average age at diagnosis of 65 years and a median overall survival (OS) of only 15 to 16 months after tumor diagnosis [4]. Long-term survival is uncommon, with fewer than 5% of patients on average surviving for five years or more after being diagnosed (source: Central Brain Tumor Registry of the United State from 2014 to 2018) [4].

Based on the new guidelines released in 2021 by the World Health Organization (WHO), GBM is classified as a grade 4 adult-type diffuse glioma based on its molecular and histopathological features. From a molecular point of view, GBM can be distinguished from other types of diffuse gliomas, such as astrocytomas and oligodendrogliomas, by its *isocitrate dehydrogenase (IDH)* wild-type status, intact chromosome arms 1p and 19q, retained expression of nuclear Alpha thalassemia/mental retardation X-linked syndrome (ATRX), and the absence of mutations in histone H3 genes. Furthermore, GBM is commonly characterized by histological features such as microvascular proliferation and necrosis, along with key molecular alterations, including the *telomerase reverse transcriptase (TERT)* promoter mutation, *epidermal growth factor receptor (EGFR)* amplification, and the +7/−10 cytogenetic signature [1].

In this review, we present a detailed overview of the current treatment options for patients with GBM, alongside an exploration of the underlying factors contributing to the failure of many anti-GBM therapies (both conventional and immune-based approaches). Furthermore, we provide an in-depth examination of the most promising immunotherapies targeting GBM, with a special emphasis on those that have already advanced to phase 3 clinical trials.

## 2. Standard of Care for GBM Patients

The established gold standard of care (SOC) for patients with newly diagnosed GBM is known as the “Stupp protocol” and comprises surgical resection, radiotherapy, and concomitant and adjuvant chemotherapy with the alkylating agent temozolomide (TMZ) [5]. If feasible, GBM interventions begin with maximal surgical resection, which eliminates most of the tumor. Surgical resection or biopsies also provide indispensable tumor material for a correct histological diagnosis and molecular testing. The extent of the tumor removed during surgery is a prognostic indicator, and according to the 2021 EANO guidelines, it should be evaluated using MRI within the first 24–48 h after the procedure [6]. Surgical resection is followed by six weeks of radiotherapy (60 Gray [Gy] in 2-Gy fractions) and concomitant daily TMZ (75 mg/m^2^), followed by six cycles of adjuvant TMZ (150–200 mg/m^2^) [5,6]. TMZ induces base methylations (i.e., N7-methylguanine, N3-methyladenine and O6-methylguanine) that, in the absence of an effective DNA damage repair system, ultimately lead to tumor cell death [7]. Of note, TMZ treatment is mostly beneficial in patients with a hypermethylated, and therefore epigenetically silent, *O6-methylguanine-DNA methyltransferase* (*MGMT*) gene. The enzyme MGMT is involved in DNA repair by removing the O6-methyl group from DNA and, if absent, enables effective chemotherapy and confers a survival advantage [5,8,9]. 

The Stupp protocol has remained unchanged over the last 18 years and typically provides an overall survival of less than two years to the patients. Thus, many clinical trials have been launched with the goal of finding new treatments to expand the life of individuals with GBM. Among various treatments, the use of tumor-treating fields (TTFs), namely low-intensity alternating electric fields delivered to the scalp of GBM individuals to induce tumor cell mitosis, has emerged as a novel modality able to ameliorate patient survival [10,11,12]. Despite the efficacy shown in a phase 3 clinical trial [11] and FDA approval, TTFs have not been yet incorporated into GBM SOC due to concerns about the unblinded nature of TTF clinical trials, as well as questions related to high cost, skin toxicity, and patient compliance [13,14].

Despite these first-line treatments, GBM virtually always recurs (median OS at recurrence = 2–9 months; median PFS at recurrence = 1.5–6 months) [15,16,17]. Once the tumor relapses, treatment options are very limited and, depending on the patient’s conditions, include second surgery, chemo-radiotherapy, and experimental treatments. As recently reviewed by Vaz-Salgato et al. (2023) [18], various second-line chemotherapeutics have been tested for the treatment of GBM, including anti-vascular endothelial growth factor (VEGF) [19,20,21], anti-transforming growth factor β (TGFβ)-receptor-I [22], anti-receptor tyrosine kinase [23], anti-protein kinase C [24], anti-EGFR [25], and anti-tyrosine kinase [26]. Although showing great promise at the preclinical level, these drugs failed to significantly improve the overall survival of GBM patients when tested in randomized clinical trials.

## 3. Therapeutic Challenges for GBM Therapies

The development of effective treatments targeting GBM could plausibly be hampered by GBM’s unique traits, including its challenging anatomical location protected by the blood–brain barrier (BBB), its invasiveness, the complexity of tumor variations within and between patients, and the immunosuppressive nature of the tumor microenvironment (TME) (Figure 1).

### 3.1. Anatomical Location

The brain is an essential organ of the human body’s governing motility, senses, emotions, cognition, memory, and survival instincts—in essence, many of the fundamental processes that regulate our body and mind. Surgical resection is therefore applicable only when GBM lies within non-critical regions of the brain that do not affect movement, speech, vision, or memory. As stated in the 2021 EANO guidelines [6], surgeons need to prioritize patients’ quality of life over extent of resection to prevent permanent neurological deficits. As recently reviewed in Bonosi et al. (2023) [27,28], there are multiple pre-operative (i.e., functional MRI imaging, magnetoencephalography, navigated transcranial magnetic stimulation, and diffusion tensor imaging) and intra-operative (i.e., ultrasonography, electrostimulation, cerebral perfusion measurements, and 5-aminolevulinic [5-ALA] tumor labeling) techniques that facilitate surgery and minimize the damages to the healthy brain tissue. As an example, patients operated with 5-ALA fluorescence-guided surgery showed a 6-month increase in progression-free survival (PFS) compared to patients operated via classical method [29,30].

### 3.2. Presence of the Blood–Brain Barrier

The brain is a highly vascularized organ and, to ensure proper neuronal functioning, needs to tightly control the trafficking of cells, molecules, and ions to and from the blood [31]. The blood–brain barrier (BBB) represents the most selective barrier of the human body, as it protects the brain from potentially harmful blood-borne agents and exogenous compounds (i.e., drugs, neurotoxins) that might damage the CNS [32,33]. It is constituted by endothelial cells of the capillaries located in the brain parenchyma, surrounded by pericytes and astrocytic endfeet, thereby isolating the brain from the bloodstream [32,34,35,36,37]. The BBB represents a major physical obstacle for the delivery of GBM therapeutics to the tumor, therefore limiting their clinical success. Indeed, a great amount of systemically administered chemotherapeutic agents failed to increase patient OS mainly due to their poor BBB penetration. An analysis of over 7000 chemotherapeutics found that only 1% of them could effectively cross the BBB and be active in the CNS [38,39]. In case of brain malignancies, including GBM, the BBB is partially disrupted leading to increased permeability, forming the so-called brain–tumor barrier (BTB). The disruption of the BBB in glioma exhibits heterogeneity, primarily manifesting within the tumor’s core while keeping its structure at the tumor rim, where invasive cells are located. The BTB stems from VEGF over-expression and increased angiogenesis in hypoxic zones, as well as the release of cytokines and chemical mediators, inducing the development of more immature and permeable vessels within the tumor [40,41,42,43,44]. Tumor-induced BBB leakage may enhance therapeutic delivery to the tumor core, yet the intact BBB beyond it can impede drug distribution. As outlined in a recent review by [45], brain drug delivery can be enhanced through surgical interventions such as intrathecal drug administration and convection-enhanced delivery (CED) and/or with the use of implantable pharmaceutical formulations, including biodegradable wafers or gels. Alternatively, researchers are focusing on improving drug penetration into the brain by enhancing drug liposolubility (e.g., using liposomes) or by modulating the BBB (e.g., through the modulation of efflux pumps, tight junctions, or the use of receptor agonists) [45]. Promising in terms of safety, these approaches require randomized clinical trials to thoroughly evaluate their effectiveness.

### 3.3. Tumor Heterogeneity and Plasticity

Another key GBM feature that can contribute to treatment failure is the high heterogeneity among (inter-tumoral) and within (intra-tumoral) tumors. Even when histologically similar, GBM tumors can differentially respond to treatments depending on their molecular profile. There are multiple signaling pathways that can be dysregulated in GBM, including p53, retinoblastoma (RB), and phosphoinositide 3-kinase (PI3K) signaling pathways [46,47]. The Cancer Genome Atlas network and subsequent studies tried to identify prognostically relevant GBM molecular subtypes based on large-scale genetic and epigenetic profiling. To date, three molecular subtypes have been proposed based on molecular analysis: proneural, mesenchymal, and classical [48,49]. They are meant to help clinicians diagnose and stratify GBM patients for potential personalized medicine [50]. However, to date, they have limited clinical relevance [51]. Moreover, researchers have recently focused on the identification of GBM subtypes by considering the characteristics and composition of the GBM tumor microenvironment. This classification system holds the potential to facilitate the implementation of precision immunotherapy approaches [52].

Inter-patient variability is further reinforced by intratumoral heterogeneity and plasticity. Within the tumor mass of an individual patient, there exists a complex, heterogeneous, and dynamic architecture of tumor cells that vary at the epigenetic, transcriptomic, protein, and metabolic levels [51,53]. Additionally, therapeutic approaches actively contribute to the phenotypic heterogeneity of GBM by modifying its tumor landscape [54]. This provides survival advantages to the tumor cells and may explain why drugs targeting the entire tumor may ultimately prove futile due to the rapid emergence of cell clones that are resistant to the specific treatment.

### 3.4. Infiltrative Nature

As with other malignant gliomas, GBM is characterized by a high invasive capacity that is associated with treatment resistance, recurrence, and poor OS. Brain tumor cells modify and degrade the extracellular matrix (ECM), enabling their invasive behavior through processes involving glutamate release and Ca^2+^ signaling pathways [55]. Within a GBM tumor, there are various levels of invasiveness reflecting the intratumoral heterogeneity of this cancer type. While tumor core cells have a higher tendency to proliferate, cells at the periphery of the tumor tend to be more invasive, allowing them to penetrate into the surrounding normal brain tissue [56]. Invasive GBM cells can move as individual cells [57] or in groups [58,59] and preferentially migrate along preexisting structures such us the brain parenchyma, white matter tracts, blood vessels, and subarachnoid spaces [60,61]. GBM cells can move along the brain tissue by remodeling the extracellular matrix and their own cytoskeleton, as well as their energy metabolism [61,62,63]. Differently from other cancer types, GBM cells rarely enter into circulation and thus do not normally metastasize to distant organs/tissues [64,65,66]. GBM cells’ invasive nature hinders complete surgical resection, and the remaining resistant clones lead to tumor recurrence [67]. As outlined in a recent review by [55], researchers have explored various approaches to inhibit invasion, including blocking Ca^2+^ channels (Mibefradil) [68], α V integrins (Cilengitide) [69], matrix metalloproteinases (MMP) [70,71], AMPA receptors (Talampanel) [72,73], and the PI3K/Akt pathway [74]. Overall, these interventions have had limited success in GBM patients.

### 3.5. Systemic and Local Immunosuppression

While historically considered “immune privileged”, the brain may be now better described as “immunologically distinct”, meaning with unique immune characteristics compared to other body parts. The brain possesses a specialized lymphatic drainage system called the “glymphatic system”, which plays a role in immunosurveillance, as it drains the cerebrospinal fluid (CSF), carrying immune cells and solutes, from the CNS into deep cervical lymph nodes [75,76]. While classical antigen presenting cells (APCs) are normally not detected in the healthy brain parenchyma, they can be found in adjacent vascular-rich tissues such as the choroid plexus and meninges [77]. They have access to the CSF and can detect brain parenchymal antigens. Moreover, in inflammatory conditions, APCs rapidly migrate towards the brain parenchyma through afferent lymphatics or endothelial venules to survey for antigens [77]. They then leave the brain and reach the deep cervical lymph nodes, where they can present brain-derived antigens and prime T and B lymphocytes, promoting adaptive immune responses [76,78]. T cells have also been observed in the brain parenchyma and CSF of healthy individuals, albeit in very low numbers, carrying out immune surveillance of the CNS and deep cervical lymph nodes [79].

As outlined in Zhang et al. (2022) [80], GBM patients often experience pronounced immunosuppression, affecting both their overall immune system (systemic) and the immune responses within the tumor environment (local). GBM patients have smaller secondary lymphoid organs and lower MHC-II expression levels in peripheral blood monocytes and are characterized by T cell lymphopenia compared to healthy individuals [81,82,83]. The decline in size and function of the thymus gland, known as thymic involution, results in decreased T cell production and, therefore, in reduced T cell availability for anti-GBM immunity [84]. T cells are majorly sequestered in the BM, due to the loss of surface sphingosine-1-phosphate receptor 1 (S1P1). S1P1 is responsible for the egress of T cells from the thymus and secondary lymphoid organs [85], but in GBM patients, the missing S1P1 receptor prevents T cells from leaving the bone marrow and entering the bloodstream [83]. Interestingly, in vitro studies revealed that serum isolated from GBM tumor-bearing mice impairs immune cell activation [86]. Circulating cytokines produced by the tumor as well as immunosuppressive treatment with corticosteroids and TMZ may further contribute to the systemic immunosuppression observed in GBM patients [81,87].

This systemic immunosuppression is further reinforced locally. In GBM, the BBB is disrupted and displays increased permeability, allowing for the influx of immune cells that are normally scarce in the brain parenchyma [88,89]. The GBM TME is highly heterogeneous and consists of various components, including GBM cancer cells, various signaling molecules, the extracellular matrix, vasculature, brain-resident non-immune cells (such as astrocytes and neurons), and lymphoid and myeloid immune cells. Despite the potential of immune responses to eliminate neoplastic cells or hinder their growth, GBM cancer cells have developed multiple mechanisms to evade immune surveillance and to shape the TME in their favor to allow for tumor development and progression. The communication between GBM cells and the TME occurs via cell-to-cell contact, soluble molecules [90,91,92], and via extracellular vesicles [93,94].

(i)Soluble molecules: Secreted by various cellular players of the GBM microenvironment, the TME contains various growth factors and cytokines, such as (i) tumor-promoting cytokines, including interleukin (IL)-1, and basic fibroblast growth factor (bFGF) and (ii) immunosuppressive chemical mediators, including TGF-β, IL-10, IL-6 and prostaglandin E-2 (PGE2) [95,96]. While IL-1 and bFGF promote tumorigenesis, TGF-β, IL-10, IL-6, and PGE2 generally shift the immune response from an inflammatory response to a pro-tumoral and wound-healing one. This alteration leads to a reduced ability of immune cells to efficiently eliminate tumor cells. Moreover, the GBM TME is characterized by high levels of CC Chemokine Ligand 2 (CCL2), a very potent chemoattractant essential for the recruitment of regulatory T cells (Tregs) and myeloid cells [97].(ii)Extracellular matrix (ECM): In GBM, ECM composition is altered due to an overexpression and increased secretion of laminin, collagen, and fibronectin, and this physically results in elevated overall density and tumor stiffness [98]. This contributes to limiting the ability of chemotherapeutic drugs to diffuse and penetrate the tumor, reducing their effectiveness. Moreover, high levels of fibronectin and hyaluronic acid, along with surrounding ECM degradation via metalloproteinases, increases the mobility and invasiveness of glioma cells [99].(iii)Vasculature: The GBM TME is characterized by abnormal vasculature, and the central areas of the tumor experience poor blood flow, leading to a decrease in oxygen delivery [100]. This hypoxic microenvironment increases the expression of hypoxia-inducible factor 1-α, promoting angiogenesis and tumor cell invasion [100]. HIF1-α upregulates immunomodulatory surface ligands such as cytotoxic T-lymphocyte-associated protein 4 (CTLA-4) and programmed death-ligand 1 (PD-L1), inhibiting efficient anti-tumor immune responses [101].(iv)Healthy brain cells: In response to CNS injury, astrocytes normally secrete growth factors and cytokines to facilitate tissue repair in a process known as astrogliosis [102]. However, in GBM, this process is exploited to promote tumor growth. In particular, the TME promotes crosstalk between astrocytes and neighboring microglia, resulting in the activation of the JAK/STAT and PD-L1 pathways within astrocytes. This activation triggers an elevated production of anti-inflammatory cytokines like IL-10, TGF-β, and STAT3, thereby fostering an immunosuppressive milieu [103]. Moreover, neurons play a role in facilitating GBM tumor progression by upregulating neuroligin-3. This leads to the activation of the PI3K signaling pathway, promoting the proliferative activity of glioma cells [104].(v)Tumor-associated myeloid cells: Tumor-associated microglia and macrophages (TAMs) are the main infiltrating population in GBM, attracted towards the tumors in response to high concentrations of various chemoattractants secreted by glioma cells, including CCL2 [105,106,107]. Within the TME, they adopt immunosuppressive and tumor-supportive phenotypes [108]. Activation of the mTOR signaling pathway leads to increased STAT3 phosphorylation and suppression of the NF-κB pathway, resulting in the upregulation of anti-inflammatory cytokines such as IL-6, and IL-10 [109]. TAMs exhibit a decreased expression of surface MHC class II molecules and costimulatory molecules (CD40, CD80, and CD86), impairing antigen presentation and activation of T cells [110,111,112]. Myeloid-derived suppressor cells (MDSCs) suppress the immune system through multiple mechanisms. They express arginase, which reduces L-arginine levels necessary for TCR expression and function. They also secrete nitric oxide and reactive oxygen species, further inhibiting T cell activity. Additionally, MDSCs express PD-L1, promoting T cell exhaustion [113,114].(vi)Tumor-infiltrating lymphocytes (TILs): In GBM, TILs often exhibit dysfunction and exhaustion caused by factors released by glioma and microenvironmental cells, including TGF-β, IL-10, and CCL2, which recruit Tregs, MDSCs, and TAMs to the tumor site [115]. In response to TGF-β, CD4+ T cells upregulate FoxP3 and differentiate into Tregs. They account for 25% of TILs and are associated with a poor prognosis in GBM [116]. Through IL-10 and TGF-β signaling, Tregs promote the transition of other T cells into regulatory ones, exert an immunosuppressive function over natural killer (NK) and CD8+ T cells, help to generate MDSCs, and impair the antigen presentation capability of DCs [117]. TGF-β1 leads to a reduction in the expression of the activating receptor natural killer group 2 (NKG2D) on the surface of both CD8+ T cells and NK cells, thereby hindering their cytotoxic effects on GBM cells [118]. Moreover, Tregs highly express immune checkpoint molecules such as PD-1 and CTLA-4 that, via interaction with their respective receptors on the surface of T cells, suppress their effector functions [119]. Glioma cells further suppress lymphocyte activity through molecules such as FasL, PD-L1, PD-L2, CD70, and ganglioside [120,121,122]. The scarcity of TILs and accumulation of exhausted T cells in the tumor microenvironment contribute to immunotherapy resistance and relapse.

## 4. Immunotherapeutic Strategies for the Treatment of GBM

Immunotherapy has revolutionized the field of oncology by aiming to re-activate the cells of the immune system to react against the tumor, rather than directly targeting the cancer cells. Immune-based approaches have shown sustained clinical benefit and, in some instances, full remission of solid tumors, thus becoming part of their standard of care [123]. However, immune-based treatments have a different impact on each cancer type depending on tumor intrinsic features and level of immunosuppression. Regarding GBM tumors, current investigations into immunotherapeutic strategies encompass immune checkpoint inhibitors, adoptive T cell therapies, vaccination approaches, and virus-based therapies (Figure 2).

Figure 2 depicts the main immunotherapeutic strategies currently under evaluation in clinical trials for the treatment of GBM. These include (i) vaccination therapy, which aims to activate the patient’s adaptive immune system via the use of tumor-specific or tumor-associated antigens, delivered in the form of nucleic acids, peptides, or packaged into DCs; (ii) adoptive T cell therapy, involving the infusion of genetically modified (chimeric antigen receptor T cells [CAR-T cells]) or activated (tumor-infiltrating lymphocytes) autologous T cells to enhance their anti-GBM activity; (iii) immune checkpoint therapy, utilizing monoclonal antibodies to remove the “brakes” on the immune system’s response to GBM; and (iv) virus-based therapy, which explores the use of viruses either to selectively target and destroy GBM cells (oncolytic viruses) or to deliver therapeutic transgenes to the tumor (cancer gene therapy). Research on combining various immunotherapies holds great promise for the treatment of GBM. The image was created with BioRender (https://www.biorender.com/, accessed on 28 February 2024).

### 4.1. Immune Checkpoint Therapy

During prolonged antigenic exposure or tumor-T cell interaction, the effector T cells might gradually lose their tumor reactivity and become “exhausted”, a hypo-responsive state characterized by high levels of co-inhibitory molecules, also known as immune checkpoints (ICMs), decreased cytotoxicity, and reduced proliferation capacity [124]. ICMs are potent regulators of the immune system exploited by the TME to suppress immune responses towards malignant GBM cells. Over the last decades, several ICMs have been identified, including programmed cell death protein 1 (PD-1) and its ligand PD-L1, CTLA-4, Lymphocyte Activation Gene-3 (LAG-3), T cell immunoreceptor with immunoglobulin and ITIM domain (TIGIT), T cell immunoglobulin and mucin domain 3 (TIM-3), V-domain Ig suppressor of T cell activation (VISTA), and indoleamine 2,3-dioxygenase (IDO).

Being surface receptors, immune checkpoints can be inhibited by blocking monoclonal antibodies, known as immune checkpoint inhibitors (ICIs). The blockade of the PD-1/PD-L1 axis or CTLA-4 have shown remarkable success in the treatment of various solid tumors, including colorectal cancer, gastric cancer, hepatocellular carcinoma, melanoma, classic Hodgkin’s lymphoma, and non-small-cell lung carcinoma [125,126,127,128,129]. However, generally, minimal clinical benefit has been observed thus far for the treatment of GBM using these modalities, whether applied individually or in combination (Table 1).

Promising preclinical results [168,169] sparked three phase 3 clinical trials testing the efficacy of the anti-PD1 antibody Nivolumab for the treatment of GBM. The first study, checkmate 143 [161], evaluated the efficacy of Nivolumab and Ipilimumab in patients with recurrent GBM. Other studies, checkmate 548 [138,170] and 498 [137], instead tested Nivolumab in addition to radiation on MGMT methylated and unmethylated newly diagnosed GBM patients, respectively. All three studies failed to achieve the primary goal of ameliorating patient survival in comparison to standard treatments. Likewise, the anti-PD1 antibody Pembrolizumab, both as a monotherapy or in combination with bevacizumab, showed limited clinical benefit for recurrent GBM patients in phase 1 [139] and 2 clinical studies [149,152,171]. It is worth noting that neoadjuvant treatment with anti-PD-1 has shown promising outcomes in selected recurrent GBM patients during window-of-opportunity trials [131,172]. Another example of immune checkpoint therapy is Durvalumab, a human IgG1 monoclonal Ab against PD-L1. PD-L1 is expressed on the surface of nearly 90% of GBM cells [173]. Radiation-induced cell death may potentiate anti-PD1 and -PD-L1 therapies by releasing tumor antigens. A phase 2 multi-center study evaluating the combination of Durvalumab and standard radiotherapy in patients with unmethylated newly diagnosed GBM demonstrated favorable tolerability and potential efficacy, with one patient achieving a remarkable OS of 86 weeks [158].

As for the FDA-approved anti-CTLA4 antibody Ipilimumab, there are currently no published clinical data available of its use as a single therapy for GBM. As GBMs can rapidly adapt to ICI therapy by increasing the expression of alternative checkpoints following treatment [174], concluded and ongoing clinical studies rather focused on the combination of Ipilimumab with other agents, including anti-PD1 blocking antibodies (NCT02311920, NCT04606316, NCT03233152, NCT04817254, NCT04145115, NCT04396860), VEGF inhibitors [175], tumor-treating fields (NCT03430791), TMZ, and radiotherapy (NCT03367715). Unlike in melanoma [176,177], combining Ipilimumab and Nivolumab in GBM yielded no added benefit and actually increased immune toxicity compared to Nivolumab alone [178].

In addition to “classical” immune checkpoints, LAG-3, TIM-3, TIGIT, and IDO1 represent novel targets that are currently under investigation in GBM. NCT02658981 and NCT03493932 phase 1 studies investigated LAG-3 blockade (Relatlimab) either as a single agent or combined with anti-PD-1 therapy in patients with recurrent GBM or newly diagnosed GBM, respectively [166,179]. Results of the treatment efficacy are awaited. NCT03961971 is currently testing the inhibition of TIM-3 (Sabatolimab) and PD-1 (Spartalizumab) together with stereotactic radiosurgery in recurrent GBM. NCT04656535 phase 0/1 study is currently recruiting recurrent GBM patients for testing the combination of Domvanalimab (targeting TIGIT) and Zimberelimab (targeting PD-1). Instead, IDO is currently under investigation in combination with other therapies (i.e., radiotherapy, TMZ) in newly diagnosed GBM patients (NCT04047706, NCT02052648) [130,164].

As recently reviewed by Arrieta et al. (2023) [180], the failure of ICI treatment in GBM can be attributed to various factors, including the low mutational burden of GBM, elevated tumor heterogeneity, limited T cell infiltration, intratumoral downregulation of MHC-I/MHC-II molecules, and insufficient drug penetration across the blood–brain barrier [112,181]. Researchers are currently focusing on combining laser interstitial thermal therapy (LITT) with ICIs, which may benefit recurrent GBM patients, as LITT ablates tumor tissue and has been shown to enhance drug penetration through the BBB breakdown [142,182,183]. Understanding of the safety and efficacy of this approach will be gained from the active ongoing NCT02311582 phase 1/2 clinical trial and from the recruiting NCT03277638 phase 1/2 clinical trial. 

### 4.2. Vaccination Therapy

Cancer vaccines represent a form of active immunotherapy that seeks to activate the patient’s adaptive immune system in response to specific antigens. These vaccines are designed to incorporate either tumor-specific antigens (TSA), also known as neoantigens, meaning mutated proteins found exclusively on tumor cells, or tumor-associated antigens (TAA), which are found to be highly expressed in the tumor but also, to a lesser extent, in normal tissues and are mostly derived from the overexpression of self-antigens [184]. Once administered, antigens are presented by APCs in the lymph nodes to naive or memory T cells. Primed T cells then migrate to the tumor site, initiating an immune response against the GBM. The objective is to trigger tumor regression and elicit durable memory responses, thereby reducing the risk of tumor recurrence. Currently, various vaccination strategies are under investigation for the treatment of GBM, employing peptides, DNA, or RNA as sources of antigens. These vaccines are packaged into various vehicles, including DCs and heat shock proteins, and are administered via intravenous, intranodal, intradermal, or intramuscular routes [184] (Table 2). To enhance vaccine effectiveness, adjuvants like tetanus toxoid, granulocyte-macrophage colony-stimulating factor (GM-CSF), and poly-ICLC (polyinosinic–polycytidylic acid stabilized with polylysine and carboxymethylcellulose) are combined with the vaccine formulation. They either promote antigen presentation, induce the expression of co-stimulatory molecules, or favor the release of cytokines [185]. 

A major challenge in vaccination strategies targeting GBM antigens is the highly heterogeneous expression of antigens within and among GBM tumors, which limits treatment response and is compounded by antigen instability and loss over time. To overcome this, the concept of a single vaccine targeting multiple antigens has been proposed to generate more robust and durable anti-tumor immune responses and reduce the risk of tumor immune evasion. However, the limited availability of neoantigens, attributed to the low mutational burden in GBM, presents a challenge in pursuing this approach [181].

#### 4.2.1. DNA/RNA Vaccines

The pioneering and extensive research by the Nobel Prize-winning Dr. Drew Weissman and Dr. Katalin Karikó on messenger RNA (mRNA) has played a pivotal role in the remarkable and swift development of mRNA-based vaccines for COVID-19. Deployed in at least 164 countries, these vaccines have been a lifeline, saving millions of lives during the global pandemic crisis, bringing considerable focus to nucleic acid vaccines in the context of cancer treatment. A notable benefit of nucleic acids is their applicability across all human HLA genotypes, enabling presentation on both MHC-I and MHC-II molecules for the activation of both CD8+ and CD4+ T cell responses [229,230].

DNA vaccines can be easily engineered, allowing for cost-effective production and purification. They also demonstrate remarkable stability and are considered safe for use. Moreover, the plasmids employed in DNA vaccines serve as potent “danger signals”, activating various DNA-sensing innate immune receptors that facilitate the development of effective adaptive immune responses [229]. However, DNA vaccines have shown a limited response in clinical trials, partly due to low in vivo transfection efficiency. By contrast, RNA vaccines provide even more advantages in terms of safety, such as the absence of risk for insertional mutagenesis, inability to self-replicate, and rapid degradation through proteases [230]. The main drawback of RNA-based therapies lies in the RNA inherent instability and limited ability to effectively penetrate cells. To increase their preservation and facilitate their delivery, RNA molecules are loaded within cells, virus-like capsid, or nanoparticles [230]. Conclusive results on the effectiveness of DNA and RNA vaccines for GBM treatment are still pending, as these vaccines have not yet undergone phase 3 clinical trials. The ongoing NCT03491683 phase 1/2 trial is investigating the combination of two DNA vaccines with a PD-1 inhibitor in newly diagnosed GBM patients. The first vaccine, named INO-5401, encodes for Wilms Tumor-1 (WT1), prostate-specific membrane antigen (PSMA), and hTERT. The second vaccine, named INO-9012, encodes for IL-12. Both vaccines are administered intramuscularly with subsequent electroporation. The latter is used as a delivery system, applying high-intensity electricity to increase membrane permeability [231]. Interim analysis shows promising results in terms of safety, immunological effectiveness, and potential survival advantage [213,232]. A phase 1 study (NCT04015700) is in progress to evaluate the efficacy of INO-9012 together with a personalized DNA vaccine, and electroporation delivery. As for RNA vaccines, a phase 1/2 study (NCT04573140) is currently investigating the intravenous administration of autologous tumor messenger RNA (mRNA) in GBM patients using lipid particles.

#### 4.2.2. Peptide Vaccines

Peptide-based vaccines use short synthetic peptides mimicking antigenic epitopes that can trigger potent and highly targeted responses. Peptide vaccines have been shown to predominantly induce humoral immunity but can also trigger cell-mediated immunity against the desired antigen [233]. So far, peptide vaccines have not demonstrated significant clinical benefit in the cure of GBM patients. This is partially due to the inherent instability and limited immunogenicity of peptides. As reviewed by Frederico et al. (2021), five main GBM-targeting peptide vaccines are currently under investigation: rindopepimut, SurVaxM, IMA950, heat shock protein–peptide complexes 96 (HSPPC-96)-specific vaccine, and personalized neoantigens vaccines [184]. Rindopepimut is a 13 aa peptide vaccine based on EGFRvIII. Despite promising results in phase 2 clinical trials [196,197,198], rindopepimut plus standard chemotherapy failed to improve the OS of newly diagnosed GBM patients in a randomized phase 3 clinical study (ACT-IV) [200]. However, trial data demonstrated increased humoral immune responses in the treatment arm compared to the control arm [200]. More than half of the trial patients, regardless of receiving rindopepimut, experienced a loss of EGFRvIII expression at relapse. This antigenic loss (~50% loss rate at relapse) reduces the number of eligible patients who can benefit from rindopepimut. Biopsy confirmation of EGFRvIII expression is therefore a crucial factor for clinical trial enrollment.

The SurVaxM vaccine specifically targets survivin, an anti-apoptotic protein that exhibits high expression in GBM while being undetectable in normal brain tissue [206,234]. Currently, a phase 2 study (NCT02455557) is actively investigating the efficacy of TMZ and the SurVaxM vaccine in treating newly diagnosed GBM patients [207,208]. Preliminary results have demonstrated the safety and tolerability of the vaccine, along with elevated levels of survivin antibodies and CD8+ T cells post-vaccination, leading to improved PFS and OS compared to historical controls [207,208]. 

The multi-peptide treatment IMA950, consisting of 11 TAAs commonly found in GBM tumors, has shown promising results in phase 1 and 2 clinical trials. Administered intradermally to newly diagnosed GBM patients treated with radiochemotherapy, IMA950 elicited CD8+ T cell responses to both single and multiple antigens [210,211]. Of note, adjuvant choice might be important for patient outcome, as the IMA950/poly-ICLC treatment [211] showed increased OS and PFS rates compared to IMA950/GM-CSF [210]. However, IMA950/poly-ICLC vaccination had no benefit in patients with recurrent GBM [212]. Phase 3 clinical trials are awaited to confirm vaccine efficacy.

Differently from IMA950, HSPPC-96 vaccine targets multiple tumor neoantigens. HSPPC-96 consists of gp96, a 96 kilodalton (kDa) heat shock protein (HSP), and its associated cellular neopeptides. As a chaperone of the ER, HSPPC-96 can be internalized into APCs for efficient antigen presentation [235,236]. Promising phase 1 and 2 results [201,202] sparked numerous clinical trials, some of which still ongoing (i.e., NCT03018288 and NCT01814813). Of note, checkpoint inhibitors may significantly impact vaccine efficacy, as an elevated PD-L1 expression translated into systemic immunosuppression and less response to vaccination [203], warranting further clinical studies on combination therapies of peptide vaccine with ICIs. Recently, researchers utilized whole-exome sequencing data to develop personalized peptide vaccines that consider the patient’s specific neoantigen expression. Phase 1 trials, including the European GAPVAC trial [223] and the American NEOVAX trial [237], have been conducted to assess the efficacy of this approach in newly diagnosed GBM patients. In both trials, the treatments stimulated robust circulating T cell responses against at least one immunizing peptide, involving CD8+ and CD4+ T cells with a memory phenotype. However, a randomized phase 3 trial evaluating personalized peptide vaccines in recurrent GBM patients did not meet the primary nor secondary endpoint for the enrolled participants [225].

#### 4.2.3. Dendritic Cell Vaccines

Another potential immunotherapeutic approach is to exploit the intrinsic antigen presentation ability of DCs to activate adaptive immune responses. Autologous DCs are typically harvested, ex vivo sensitized with antigens and then re-infused into the patient [238,239]. Autologous DCs can be directly isolated from the peripheral blood or differentiated in vitro from monocytes or CD34+ hematopoietic stem cells via IL-4 or GM-CSF [240]. DCs may be “educated” via several forms of antigens ranging from DNA/RNA to peptides and tumor lysates. Peptides loaded on DCs are more efficiently delivered to the target tissue compared to peptide treatments alone. Although the initial clinical results appear promising, there is currently a scarcity of robust evidence regarding the efficacy of DC vaccines in GBM. The outcome of DC vaccines against GBM tumors is variable, reflecting inter-individual heterogeneity and ranging from minimal or no clinical response to significant response. Additionally, without the aid of adjuvants, DCs face challenges in migrating to the lymph nodes, with less than 5% of injected DCs successfully reaching their target destination [241].

CMV proteins are highly expressed in over 90% of GBM tumors but are rarely found in healthy brain tissue [242]. mRNA encoding for the CMV phosphoprotein 65 (pp65) can be loaded into DCs to stimulate CMV-specific T cell immunity able to kill GBM cells [243]. Two phase 1 studies [189,190] demonstrated that, despite the cold microenvironment of GBM, CMV-pp65 RNA-pulsed DCs (also known as CMV-DCs) triggered antigen-specific T cell responses, warranting further follow-up (NCT02771301, NCT02465268). The pre-conditioning of patients with tetanus/diphtheria toxoid actively increased the homing of pp65-specific DCs to the lymph nodes [189].

To date, only two DC vaccines reached randomized phase 3 clinical trials: ICT-107 and DCVax-L. In ICT-107, DCs are pulsed with multiple MHC-I-restricted TAAs highly expressed on GBM: AIM-2, MAGE-1, HER2/neu, TRP-2, gp100, and IL-13Rα2 [244,245]. A phase 2 study demonstrated the safety and immunogenicity of the treatment, as well as an improvement in patients’ PFS compared to the control group [209]. The phase 3 trial (NCT02546102) testing the intradermal administration of ICT-107 in newly diagnosed GBM patients was prematurely suspended because the company was unable to financially support its completion.

For DCVax-L, DCs are pulsed ex vivo with a tumor lysate. In a randomized phase 3 clinical trial, the effectiveness of DCVax-L and standard radiochemotherapy was evaluated in patients with newly diagnosed and recurrent GBM. The addition of DCVax-L to the standard therapy was found to be safe [220]. The multicentric study (NCT00045968) started in 2007 over a period of eight years and included two arms of GBM patients. In addition to standard radiochemotherapy, the first arm was treated with placebo, while the second arm received DCVax-L. The primary endpoint of the trial was PFS. However, in the initial report detailing the trial results, there was no mention of PFS data. Instead, the authors declared an increase in OS [220]. After four years, a second report retrospectively compared the OS of DCVax-L-treated patients with that of an external control population of patients obtained from selected published randomized clinical trials [221]. The data suggested that MGMT-methylated patients show increased survival compared to non-methylated individuals, pointing to a possible cooperative effect of TMZ and DCVax-L. Notably, the treatment led to an extension of median OS for both newly diagnosed GBM (19.3 months vs. 16.5 months) and recurrent GBM (13.2 months vs. 7.8 months) patients compared to external controls receiving standard of care alone [221]. However, concerns were raised regarding the interpretation of the results, emphasizing the necessity to approach the findings with caution. Various design issues, such as a shift in the primary endpoint from PFS to OS based on arguments related to pseudo-progression, an extended duration of the enrollment period, and an inappropriate selection of the control arm, contribute to these concerns [246,247,248,249,250,251].

### 4.3. Adoptive T Cell Therapy

Adoptive T cell therapy is an immunotherapy technique in which the patient’s T cells are expanded outside the body (ex vivo) and then reinfused back into the patient to target tumors. A few days before T cell reinfusion, patients undergo a lymphodepleting preparative regimen, which involves the use of lymphocyte-directed chemotherapy. This regimen aims to create a favorable environment that prolongs the persistence of infused cells and enhances the effectiveness of the treatment [252]. Currently, adoptive T cell therapy in the context of GBM primarily involves the use of patient-isolated infiltrating T cells (TIL therapy) or patient-isolated T cells genetically engineered ex vivo to regain cancer-fighting properties, such as chimeric antigen receptor T cells (CAR-T cells) (Table 3).

#### 4.3.1. TIL Therapy

The preparation of autologous TILs is a time-consuming process with a low success rate. It involves culturing a resected tumor specimen in a high concentration of recombinant IL-2, along with IL-15 and IL-21 if necessary. The TILs are then selected, expanded, and transferred to the patient. A pilot study demonstrated that the delivery of autologous TILs and IL-2 had limited anti-tumor effects in the context of malignant gliomas [265]. As a potential explanation, patient-isolated TILs are heterogeneous in terms of TCR and level of exhaustion and would therefore react differently against the tumor cells [83,266]. The use of ICIs may therefore promote the anti-tumor efficacy of TIL therapy. Two phase 1 clinical trials (NCT05333588, NCT04943913) are currently recruiting GBM patients to investigate safety of TIL therapy, with results expected for 2024–2025.

#### 4.3.2. CAR-T Cell Therapy

A promising T-cell-based approach involves the genetic engineering of autologous T cells to express a chimeric antigen receptor (CAR) designed to target tumor-specific antigens. CAR is a recombinant receptor that, in its latest generations, consists of four main components: (i) an extracellular antigen-recognition domain, (ii) a spacer region, (iii) a transmembrane domain that anchors CAR to the cell membrane, and (iv) intracellular signaling domains that provide co-stimulation and initiate the signaling cascade [267]. The major advantage of CAR-T cell therapy is that CAR recognizes a tumor antigen independently of MHC-restriction, therefore bypassing antigen presentation. Once bound to a specific antigen, the CAR signaling domains send the signals to the T cell to kill the target cell. 

Driven by the success of CAR-T therapies in hematological cancers [268], researchers are currently focusing their efforts on the development of GBM-specific CAR-T therapies. So far, CAR-T cell clinical trials for GBM are still in the early stages, primarily in phase 1/2 trials. While some CAR-T cells have shown promise, they still need to demonstrate clinical benefits conclusively. The interpatient variability in surface antigen expression along with the problem of antigen escape represent major obstacles of this approach. Other barriers to the clinical efficacy of CAR-T cells are T cell engraftment and expansion in vivo and the inhibitory TME, which becomes even more immunosuppressive after CAR-T therapy [269]. Combining lymphodepleting preconditioning and ICIs may address these obstacles. Moreover, the high cost of CAR-T cell manufacturing can affect healthcare expenditures and limit access to this therapy. As reviewed by Luksik et al. (2023), EGFRvIII, IL-13Rα2, and HER2/neu are among the main target antigens of CAR-T cell therapy evaluated in clinics in the last decade [270]. B7-Homolog 3 (B7-H3), the ECM metalloproteinase inducer (EMMPRIN), disialoganglioside (GD2), matrix metalloproteinase 2 (MMP2), and NKGD2 ligands are instead novel targets currently under investigation in ongoing clinical trials [271].

EGFRvIII-directed CAR-T cells were tested in a phase 1 study for the treatment of EGFRvIII+ recurrent GBM, showing safety and feasibility without cross-reactivity to wild-type EGFR. However, the therapy resulted in EGFRvIII antigen escape and adaptive resistance [253]. A subsequent phase 1/2 trial did not yield clinical benefits in recurrent GBM patients [257].

IL-13Rα2 is a potential target found in many human cancers, including GBM (>75%) [272]. Different versions of IL-13Rα2-targeted CARs have been developed so far, with modifications in genetic elements and costimulatory molecules [260,261,273,274]. 13Rα2-targeted CAR-T cells showed promising results in a recurrent GBM patient, with tumor regression, increased cytokine levels, and no therapy-associated toxicity. The clinical response lasted for 7.5 months after treatment [260]. The newest version of IL-13Rα2-targeted CAR-T cells was genetically modified to induce a permanent disruption of the glucocorticoid receptor. In a phase 1 trial, the intracranial administration of the therapy in recurrent GBM patients was well tolerated, with indications of transient tumor reduction and/or tumor necrosis at the site of T cell infusion [274].

HER2/neu, being overexpressed in 80% of GBM, is another common antigenic target used in CAR-T therapies [275]. Despite its expression in both tumor and healthy brain tissue, no off-target toxicity has been observed in GBM patients systemically infused with HER2/neu-specific CAR-T cells [276]. Hedge and colleagues designed and created bivalent HER2/neu- and IL-13Rα2-targeting CAR-T cells that, in preclinical GBM mouse models, reduced antigen escape, enhanced T cell effector functions, and improved animal survival [277]. Trivalent CAR molecules specific for the glioma antigens HER2/neu, IL-13Rα2, and ephrin-A2 (EphA2) have the potential to capture nearly the totality of tumor cells. In preclinical models, these CAR-T cells inhibited tumor growth and extended animal survival compared to monospecific or bispecific CAR-T cells [278]. Clinical trials are still awaited to confirm treatment efficacy in humans.

### 4.4. Virus-Based Therapy

Virus-based treatments employed for the treatment of GBM can be either gene delivery systems or oncolytic viruses (OVs) (Figure 3, Table 4). Viral vectors are non-lytic and typically deliver pro-inflammatory and anti-angiogenic molecules, tumor suppressor genes, TAAs, ICIs, small interfering RNAs, cancer stroma-degrading enzymes, and cytotoxic convertases [279]. OVs are instead replication-competent viruses that selectively replicate in cancer cells inducing their lysiswhile sparing the heathy counterparts. They can either have inherent oncolytic properties by naturally infecting tumor cells or acquire selective tropism through genetic modifications [280]. Due to their replicative nature, OVs induce cell lysis, which in turn elicits secondary immune responses by releasing viral PAMPs, DAMPs, and TAAs. The infection of tumor cells with OVs has the effect of “warming up” the immunosuppressive TME, resulting in the inhibition of tumor progression and an enhanced suitability of the TME for other therapeutic interventions [280].

#### 4.4.1. Adenovirus (AdV)

In the context of GBM, researchers have primarily focused on the AdV delivery of the herpes simplex virus (HSV) *Thymidine kinase* (*TK*) gene, the *TP53* tumor suppressor gene, the IL-12-encoding gene, and a transgene encoding for a chimeric death receptor (VB-111). 

When administered alongside ganciclovir or valacyclovir, HSV-*TK* converts them into cytotoxic products that accumulate and selectively eliminate the transduced cancer cells. The various clinical trials testing HSV-*TK*/ganciclovir gene therapy differed in the promoter used to control *TK* gene expression: (i) Rous sarcoma virus (RSV) promoter [288,318,319] and (ii) CMV promoter [320,321,322]. AdV-mediated gene therapy was safe and well tolerated [318,319,320]. A phase 2 trial testing the infusion of the suicide gene therapy into the arteries in patients with recurrent GBM revealed an improvement of PFS (29.6 vs. 8.4 weeks) and OS (45.4 vs. 14.3 weeks) compared to standard treatments alone [288]. In a phase 3 randomized, controlled study by Immonen et al. (newly diagnosed GBM and recurrent GBM patients), HSV-*TK* showed little to moderate improvement in survival rates and moderate tolerability [321,322]. The substitution of ganciclovir with valacyclovir was found to be safe [287] and resulted in improved median OS (17.1 vs. 13.5 months) for newly diagnosed GBM patients compared to standard treatments alone, as observed in a phase 2 study [289].

A second genetic approach used for GBM treatment consists of the upregulation of the *TP53* tumor suppressor gene [323]. Restoration of a functionally active p53 protein was achieved via the use of a *TP53*-armed AdV (INGN 201; ADVEXIN) constructed through cDNA of the wild-type *TP53* in place of the AdV *E1* region [324]. The virus showed minimal cytotoxicity in vivo but, when intratumorally injected, failed to distribute widely in the tumor tissue, reaching only 5 mm from the injection site. Most notably, one GBM patient enrolled in the clinical study survived nearly 3.5 years after Ad-*TP53* treatment without evidence of recurrence [325]. The *p53*-armed AdV was also investigated in another phase 1 clinical trial (NCT00004080), but the results are not yet available.

Researchers investigated the effects of the proinflammatory cytokine IL-12 on GBM tumors using an engineered AdV-based vector called Ad-RTS-IL-12 [326]. This vector allows for the inducible expression of IL-12, activated via oral administration of veledimex. Preclinical studies showed reduced tumor mass and increased lymphocyte infiltration [326]. In human application, Ad-RTS-IL-12 is injected into the resection cavity of recurrent GBM patients, accompanied by veledimex administration, showing limited toxicity and promising anti-tumor immune responses [290].

Lastly, VB-111 is an AdV-based cancer gene therapy that specifically targets angiogenic endothelial cells with a transgene encoding a chimeric death receptor, linking Fas to human TNF-R. When activated, this receptor induces Fas-mediated apoptosis and vascular disruption, leading to tumor starvation. In a phase 2 study, the combination of VB-111 and bevacizumab doubled the survival of patients with recurrent GBM compared to bevacizumab monotherapy [291]. However, a randomized controlled phase 3 study (GLOBE), testing VB-111 and bevacizumab failed to replicate the phase 2 results in recurrent GBM patients [292]. 

Alternatively, researchers have tested oncolytic AdVs, also known as conditionally replicative adenoviruses (CRAds), to target GBM tumors. These viruses acquire their tumor specificity via either (i) deletion of genes encoding for cell cycle regulatory proteins, (ii) natural overexpression of virus receptors on the surface of tumor cells, or (iii) use of tumor-specific promoters to control viral replication [327]. In the case of GBM, four main CRAds have reached clinical testing: ONYX-015, DNX-2401, DNX-2440, and CRAd-S-pk7.

ONYX-015 contains a deletion of the *E1B* gene. The virus preferentially replicates in cancer cells through various, not yet fully characterized mechanisms [328,329]. At the preclinical level, ONYX-015 achieved promising results in terms of tumor cell killing and reduction of tumor mass [330]. In a phase 1 study, ONYX-015 proved to be safe and well tolerated even at the highest dose (10^10^ viral particles) in all enrolled patients, among which recurrent GBM cases were included [285]. However, no tendency of anti-tumor efficacy could be observed in this study [285]. 

DNX-2401, previously known as delta-24-RGD (Δ24RGD) or Tasadenoturev, features a 24 bp deletion of the *E1A* gene that abrogates the binding and inhibition of E1A to the Rb protein and a fiber knob RGD modification to retarget virus entry via cell surface integrins that are typically enriched in glioma cells. These modifications were initially believed to enable selective targeting and replication of the virus to cancer cells with aberrant Rb pathways [331,332]. However, other research groups have been unable to replicate these initial findings [333]. Both as a single agent or in combination with other treatments (i.e., IFN-γ and anti-PD1), DNX-2401 did not raise any safety concerns [283,334,335,336]. Although the 12-month survival objective was achieved, the combination of DNX-2401 with TMZ and pembrolizumab did not meet the primary endpoint of objective response in a phase 2 clinical trial [337]. A new clinical trial (NCT03896568) is actively recruiting recurrent GBM patients to test DNX-2401 oncolytic virus delivered by allogenic bone marrow-derived human mesenchymal stem cells.

DNX-2401 has been recently modified to express the human OX40 co-stimulatory ligand (OX40L), aiming to enhance the antigen presentation in tumor cells. Compared to DNX-2401, this new version exhibited more potent and specific anti-glioma activity, attributed to superior T cell activation and proliferation [338]. Although a phase 1 clinical trial (NCT03714334) was underway to evaluate this modified virus for recurrent GBM treatment, it was terminated due to a stock shortage.

Lastly, Ulasov and colleagues generated a glioma-specific recombinant AdV, called CRAd-S-pk7, by modifying the Ad5 fiber with pk7s and by regulating the expression of the *E1A* gene via the human survivin promoter [339]. Building on encouraging preclinical results [340,341], CRAd-S-pk7 virus loaded onto neural stem cells was administered during surgery in newly diagnosed GBM patients, along with chemo-radiotherapy [281]. The treatment proved to be safe and well tolerated [281]. Although not the primary objective of the study, the presence of promising survival outcomes provides support for conducting further investigations of CRAd-S-pk7 in phase 2/3 clinical trials.

#### 4.4.2. Retrovirus

In the context of GBM, researchers have primarily focused on the retrovirus delivery of the HSV-*TK* gene, or of the yeast cytosine deaminase gene (Toca 511). A phase 3 study that tested HSV-*TK* gene delivery along with intravenous ganciclovir administration demonstrated no significant differences in median OS between treatment and control patients [302]. Toca 511, also known as Vocimagene Amiretrorepvec, is a replication-deficient engineered murine leukemia virus armed with the yeast *cytosine deaminase* gene [342]. When administered in combination with the prodrug 5-fluorocytosine (Toca FC or 5-FC), the virus-delivered cytosine deaminase converts it into its toxic form 5-Fluorouracil (5-FU) that eventually kills the cancer cells and nearby immunosuppressive cells [343]. Of note, 5-FU can induce so-called “bystander effects”, as it can passively diffuse through cell membranes, therefore not only affecting directly infected cancer cells but also nearby cancer cells [344]. Despite encouraging observations in a phase 1 study [306], similarly to the case of ICIs, Toca 511/5-FC failed to meet the primary endpoint of improve patient survival compared to standard of care when tested in a randomized open label phase 2/3 study [308].

#### 4.4.3. Herpes Simplex Virus (HSV)

The neurotropic HSV-1 belongs to the *Herpesviridae* family, and it is an enveloped icosahedral virus with a dsDNA genome. To date, three genetically engineered versions of it have been evaluated in completed clinical trials: HSV-1716 [345], G207 [298,346], and G47Δ [347]. First-generation HSV-1716 contains a deletion of *γ134.5* genetic loci, which counteracts the normal antiviral response of cells and allows viral protein translation to proceed [345]. Three UK phase 1 clinical trials demonstrated the safety of intratumoral injection of it, either alone or following surgical resection, in glioma patients [293,294,295]. The second-generation G207, which includes an additional insertion of the *UL39* gene preventing viral replication in non-dividing cells [298,346], also demonstrated safety [296,297,298]. The third-generation G47Δ (Teserpaturev, DELY-TACT) differs from the G207 backbone for a *α47* gene deletion that enhances viral replication and triggers anti-tumor immune-mediated responses via upregulation of MHC-I molecules [347]. Of note, the G47Δ bears the same genetic mutations (*γ134.5* and *α47*) of the first FDA- and EMA-approved oncolytic virus, namely T-VEC (Talimogene Laherparepvec; IMLYGIC^®^; formerly called OncoVEX^GM-CSF^) [348]. However, via additional deletion of *UL39*, G47Δ was more attenuated than T-VEC and therefore safer. At the University of Tokyo, a phase 1/2 study demonstrated the safety of G47Δ when intratumorally injected in recurrent GBM patients [299]. Accordingly, they started the subsequent phase 2 study to test the efficacy of multiple intratumoral G47Δ injections (1 × 10^9^ viral particles; max of six injections) in patients with recurrent GBM [299]. Based on outstanding clinical results, G47Δ received a conditional time-limited approval by the Pharmaceuticals and Medical Devices Agency of Japan (PMDA) for the treatment of brain tumors. 

#### 4.4.4. Poliovirus

Polioviruses are positive single-stranded RNA (ssRNA) viruses belonging to the *Picornaviridae* family. PVSRIPO, or Lerapolturev, is a non-pathogenic poliovirus/rhinovirus chimeric virus with anti-neoplastic activity [349]. PVSRIPO specifically targets tumor cells by utilizing the poliovirus receptor CD155 [350]. In a phase 1 trial, intratumoral treatment with PVSRIPO in recurrent GBM patients demonstrated an improved overall survival compared to historical controls [314]. Ongoing clinical studies include a phase 2 trial (NCT02986178) investigating PVSRIPO as monotherapy, as well as phase 1/2 (NCT03973879) and phase 2 (NCT04479241) trials exploring the combination of PVSRIPO with either anti-PD-L1 atezolizumab or anti-PD1 pembrolizumab, respectively.

#### 4.4.5. Respiratory Enteric Orphan Virus (Reovirus)

Reoviruses are naturally occurring double-stranded RNA viruses that belong to the *Reoviridae* family. They are non-pathogenic and selectively replicate within cancer cells by taking advantage of the Ras pathway that is commonly upregulated in neoplastic cells [351]. They underwent four phase 1 clinical trials for GBM treatment, with each study exploring a different administration route: intratumoral [315,316] or systemic [317] injection. In all trials, Reolysin proved to be safe. Of note, treatment causes an in vivo upregulation of IFN-regulated genes and PD-1/PDL-1 axis, as well as an increase in T cell infiltration [317]. This makes Reolysin particularly interesting for combination therapies.

#### 4.4.6. Measles Virus (MeV)

MEVs belong to the *Paramixoviridae* family and contain a negative sense ssRNA genome. They were originally chosen to treat malignancies, as a case report linked their infection to tumor remission [352]. The virus used for GBM treatment is a live attenuated strain called MV-CEA that preferentially enters and replicates within malignant cells, including GBM [353]. MV-CEA demonstrated to be safe in an early phase 1 trial testing the injection of the virus in the tumor resection cavity of recurrent GBM (NCT00390299).

#### 4.4.7. Newcastle Disease Virus (NDV)

NDV is an avian paramyxovirus with intrinsic oncolytic potential [354]. It is a negative-sense ssRNA virus that preferentially replicates within type I IFN-deficient cancer cells [355]. The HUJ [310] and MTH-68/H [311] strains of NDV have been the subject of clinical studies in patients with recurrent GBM. A phase 1/2 study of systemic application of NDV-HUJ revealed minimal toxicity and encouraging anti-tumor responses, with one patient achieving complete tumor remission during maintenance dosing [310]. However, the complete response was not durable.

#### 4.4.8. H-1 Parvovirus (H-1PV)

Another promising strategy in the fight against GBM is the use of the oncolytic H-1PV. It is a rat protoparvovirus of the *Parvoviridae* family characterized by an ssDNA genome. It is not pathogenic for humans and naturally possesses oncolytic and oncosuppressive properties as demonstrated in various in vitro and in vivo models [356,357]. Wild-type H-1PV treatment was successful in a phase 1/2 clinical trial for recurrent or progressive GBM, where patients received initial H-1PV administration via intravenous or intratumoral injection, followed by surgical resection and virus re-injection into the resection cavity [312]. Results show that the treatment is safe, well tolerated, and associated with surrogate evidence of efficacy, including immune conversion of the TME and extended patient median OS in comparison with historical controls [312,313]. Compassionate use programs explored the combination of H-1PV with different agents, particularly bevacizumab, an anti-angiogenic agent with immunomodulating properties [358], the PD-1 inhibitor Nivolumab, and alongside Valproic acid, owing to encouraging preclinical results [359]. This multimodal therapeutic approach led to partial or complete objective responses in seven out of nine cases [360,361]. 

#### 4.4.9. Vaccinia Virus (VACV)

Enveloped dsDNA vaccinia viruses belong to the *Poxviridae* family and, in most of cases, harbor inactivating mutations of the TK-encoding *J2R* gene (ΔJ2R VACV). ΔJ2R VACV therefore depends on host cells for TK protein, which is overexpressed in tumor cells [362]. Researchers developed the virus TG6002 by engineering a ΔJ2R VACV Copenhagen strain to express the yeast suicide gene *FCU1* [363]. When combined with 5-FC, TG6002 activates the prodrug, leading to tumor cell death by inhibiting DNA and protein synthesis. A concluded Phase 1 trial (NCT03294486; ONCOVIRAC) tested the safety and efficacy of TG6002/5-FC in recurrent GBM patients; however, the results are not yet posted. 

## 5. Combination Therapy

It has become increasingly evident that a singular treatment approach is insufficient for effectively addressing tumors, especially when dealing with a complex and heterogeneous entity like GBM. Researchers are now directing their attention toward combination therapies, seeking not only to combine immunotherapeutics with conventional treatments but also to explore synergies among different immune-based approaches (Table 5). 

Immune checkpoint inhibitors are currently being tested in combination with CAR-T cells therapies (NCT03726515, NCT04003649), vaccination approaches (NCT03422094, NCT02287428, NCT04013672, NCT03014804, NCT04201873), and with oncolytic viruses such as AdVs (NCT03576612, NCT03636477), HSV (NCT05084430, NCT02798406), and PVSRIPO (NCT04479241, NCT03973879).

In addition to exploring immunotherapeutic strategies, it is crucial to consider the integration of radiation therapy into the treatment landscape for GBM. Being a first-line treatment and integral component of the Stupp protocol, combining radiation with immunotherapy is a logical approach. However, this combination introduces both opportunities and challenges. On the one hand, radiotherapy, with its tumor-targeting ionizing radiations, induces molecular lesions, including DNA breaks (single- and double-stranded) and base modifications triggering immunogenic cell death [369]. As extensively reviewed in De Martino et al. (2021) [370], radiotherapy has the potential to enhance GBM sensitivity to immune-based approaches by actively recruiting effector T cells to the tumor site, an essential requirement for successful immunotherapy. However, the intricate interplay between radiation and immune therapies demands careful consideration, as certain aspects of radiation might counteract immunotherapeutic mechanisms [369]. For instance, B cells, T cells, and NK cells are among the most radiosensitive cells of the TME, while immunosuppressive Tregs and MDSCs are quite resistant to radiation. The combination of radiotherapy with various forms of immunotherapy is an active area of research, with experiments in animal models demonstrating its potential efficacy and benefits. Building on these promising preclinical data, some clinical trials are strategically combining specific types of radiation therapy with immunotherapeutic to harness potential synergies [369]. Understanding the nuances of how radiation influences the immune response is essential for optimizing treatment outcomes and advancing the development of effective combination therapies for GBM.

## 6. Conclusions and Future Directions

GBM patients’ poor prognoses underscore the urgent need for novel treatments to enhance both the quality of life and overall survival for patients. While immunotherapeutic approaches have demonstrated significant efficacy in treating solid tumors, their effectiveness in addressing GBM remains limited. Despite promising results at the preclinical level, anti-GBM immunotherapeutics, whether tested individually or in combination with standard treatments, have so far failed to yield clinically meaningful outcomes when examined in phase 3 clinical trials. This high failure rate highlights the pressing need for more reliable preclinical models and early-stage clinical studies. Moreover, a better understanding of GBM tumor biology, in terms of local TME immunosuppression and systemic T cell dysfunction, is essential in the development of more targeted therapies. Recent advances in patient-derived GBM xenografts in humanized and immunotolerant murine models, as well as in ex vivo 3-D systems and microfluidics, can assist researchers in studying the intricate relationship between GBM and immune cells, leading to the discovery of new ways to efficiently modulate it [371]. Furthermore, these models serve as excellent preclinical settings for the high-throughput screening of therapeutic agents in a time-efficient and cost-effective manner. Artificial intelligence and machine learning can enhance preclinical models, supporting research efforts, and accelerating relevant discoveries. On the clinical side, the majority of phase 2 GBM studies are currently conducted in single-center settings with single-arm designs. A shift towards randomized, controlled, and adequately powered clinical studies can significantly contribute to the development of more robust therapies, preventing the wastage of valuable patient and financial resources and maximizing the reproducibility of results. Clinical trials should also consider including immune-predictive biomarkers and genomic characterization of tumors. This information could provide the key towards more personalized therapies addressing specific tumor signatures and are active areas of intense research.

Standard chemoradiotherapy is well-known for inducing immunosuppression and lymphopenia in GBM patients, posing a significant obstacle to GBM immune-based approaches. Essential changes in current standard treatments are required to increase the success of immunotherapies [372]. Moreover, failed clinical trials have taught us that targeting a single axis, such as a single antigen or immune checkpoint molecule, may not lead to success. Antigens exhibit heterogeneous spatial and temporal expression within tumors, influenced by the tumor microenvironment, treatment, tumor progression, and environmental factors. Consequently, CAR-T therapies are now simultaneously targeting three (trivalent) or more (polyvalent) antigenic targets, and peptide/DC vaccines increasingly utilize the entire tumor lysate rather than a single tumor antigen. Moreover, bispecific T cell engagers (BiTEs), which physically brings T cells in close proximity to tumor cells, have been proposed as a possible solution to overcome antigen escape mechanisms [373]. In addition, various personalized immune-based treatments, customized to individual patient profiles, are currently undergoing clinical evaluation and may hold the key to addressing the challenges posed by GBM. Neoantigen-based personalized vaccines demonstrate significant immunogenicity and safety in GBM, generating robust CD8+ and CD4+ T cell infiltration into the tumor. Alongside the personalization aspect is the consideration of combination therapy; it is crucial to comprehend which therapies synergize effectively and, notably, to determine the optimal timing for their administration to achieve maximum results.

The high costs associated with immunotherapies for GBM, especially in the realm of combination therapies, underscore the pressing need for sustainability in their pricing. To achieve this, stakeholders should focus on increasing research funding, fostering collaborative efforts, implementing regulatory incentives, and promoting value-based pricing. Additionally, encouraging global health partnerships, supporting insurance and health system reforms, and establishing patient assistance programs are crucial steps towards making these treatments more accessible and averting potential healthcare system collapses. By addressing these challenges, we can also work towards mitigating inequalities in access to GBM treatments, ensuring that all patients, regardless of their financial status, have equitable access to life-saving therapies.

## Figures and Tables

**Figure 1 cancers-16-01276-f001:**
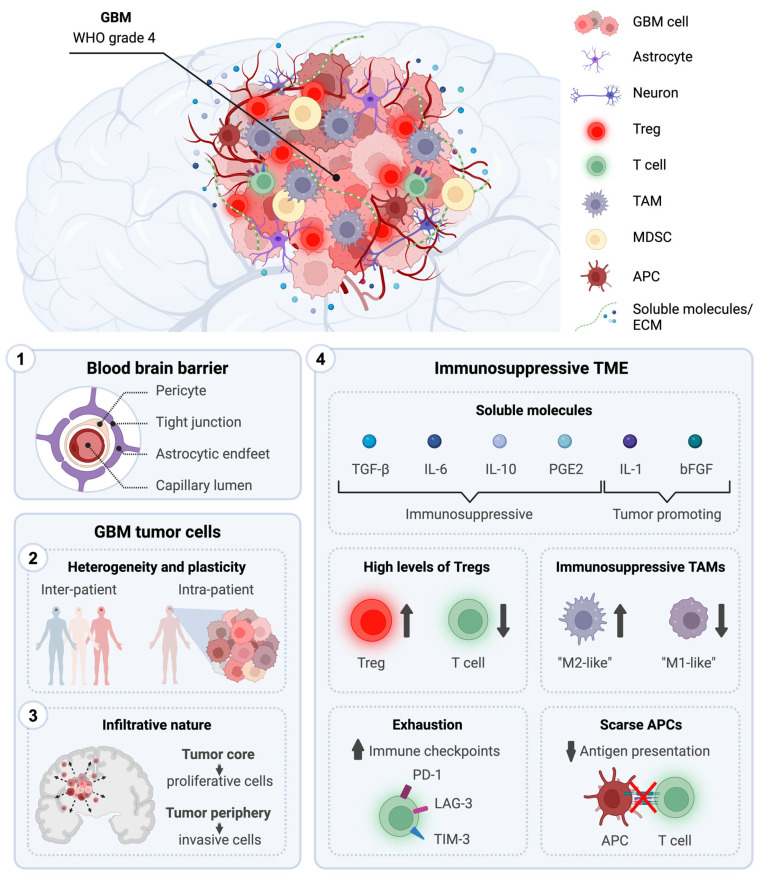
Therapeutic challenges for the cure of GBM. Abbreviations: APC, antigen-presenting cell; bFGF, basic fibroblast growth factor; ECM, extracellular matrix; GBM, glioblastoma; IL, interleukin; LAG-3, lymphocyte-activation gene 3; MDSC, myeloid-derived suppressor cells; PD-1, programmed cell death protein 1; PGE2, prostaglandin E2; TAM, tumor-associated microglia and macrophages; TGF-β, transforming growth factor-β; TIM-3, T-cell immunoglobulin and mucin domain; TME, tumor microenvironment; Treg, regulatory T cell; WHO, the World Health Organization. The figure illustrates the distinctive characteristics of GBM (WHO grade 4) that are understood to hinder the development of effective anti-tumor therapies. These include (**1**) an anatomical location shielded by the blood–brain barrier, (**2**) intra- and inter-patient tumor heterogeneity, (**3**) infiltrative behavior, and (**4**) a highly immunosuppressive TME. The latter showcases the presence of GBM-driven cytokines with immunosuppressive and tumor-promoting properties, along with immunosuppressive cell populations such as Tregs and M1-like TAMs, accompanied by upregulated exhaustion markers. Additionally, GBMs strategically downregulate antigen-processing and presentation molecules to effectively evade T cell activation. The image was created using BioRender (https://www.biorender.com/, accessed on 18 December 2023).

**Figure 2 cancers-16-01276-f002:**
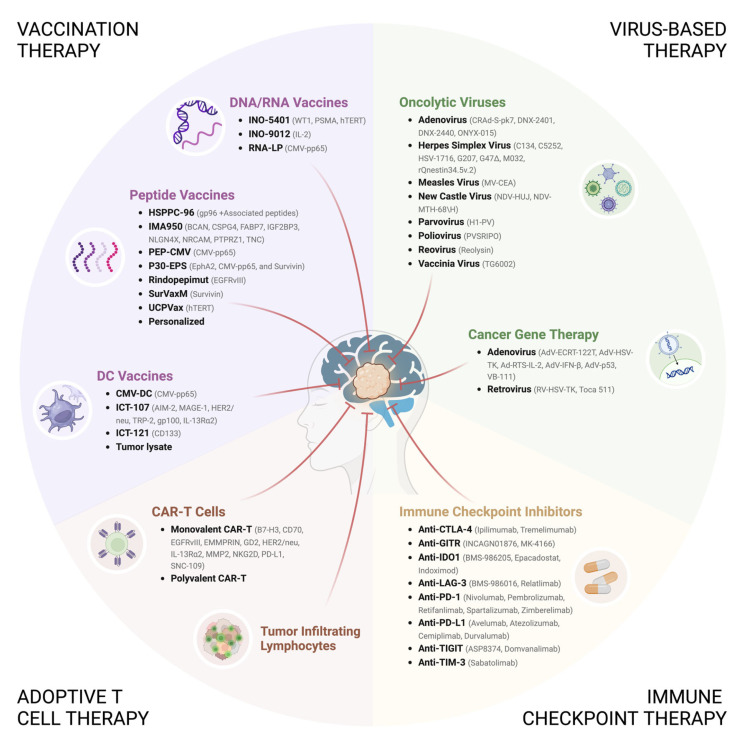
Overview of the main immunotherapeutic modalities against GBM.

**Figure 3 cancers-16-01276-f003:**
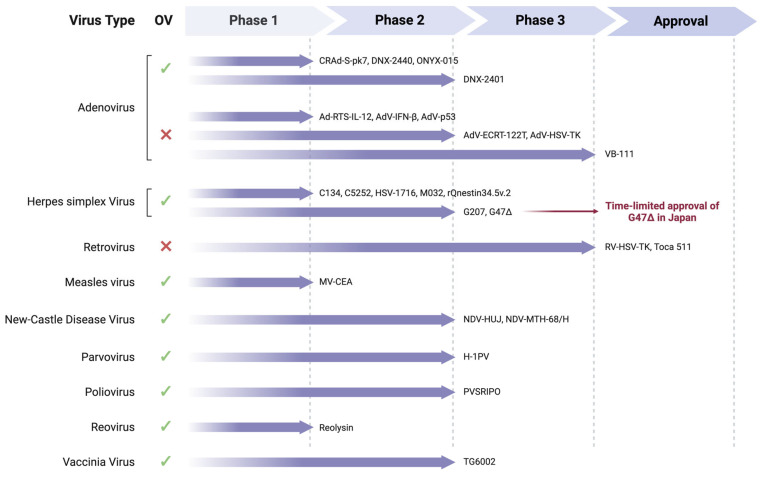
Past and ongoing clinical trials in virus-based therapies for GBM. This figure provides a comprehensive overview of the clinical studies investigating oncolytic virus (OV) or non-lytic viral vectors for the treatment of GBM. A check mark under the “OV” section signifies the virus is oncolytic, while a cross mark indicates its use as a non-lytic viral vector. The image was created with BioRender (https://www.biorender.com/, accessed on 18 December 2023).

**Table 1 cancers-16-01276-t001:** List of clinical trials involving ICIs in adult GBM patients. The table includes concluded or terminated studies, as well as those currently underway or preparing to enroll participants. Data were sourced from Clinicaltrials.Gov, retrieved on 13 December 2023.

Inhibitor	NCT Number	Phase	Study Status	Tumor Target	Intervention	Outcome
Anti-CTLA-4 (Ipilimumab)	NCT05074992	2	Terminated	ndGBM	Ipi	
Anti-IDO1 (Indoximod)	NCT02052648 [130]	1/2	Completed	Malignant Brain Tumors	IND + TMZ	
					IND + TMZ + Bev	
					IND + TMZ + Stereotactic RT	
Anti-PD-1 (Nivolumab)	NCT02648633	1	Terminated	rGBM	Valproate + Stereotactic RT + Nivo	
	NCT02550249 [131]	2	Completed	GBM	Neoadjuvant Nivo	mOS: 7.3 months (95% CI, 5.4–7.9), mPFS: 4.1 months (95% CI, 2.8–5.5)
	NCT02335918 [132]	2	Completed	Advanced Solid Tumors	Nivo + Varlilumab	OS-12: 40.9%
	NCT03890952 [133]	2	Active Not Recruiting	rGBM	Nivo + Bev + Surgery	
					Nivo + Bev	
	NCT04195139 [134]	2	Active Not Recruiting	ndGBM	RT + TMZ + Nivo	mOS: 11.8 months, PFS-6: 64%
					RT + TMZ	mOS: 12.0 months, PFS-6: 49%
	NCT03743662	2	Active Not Recruiting	rGBM (MGMT-M)	RT + Bev + Nivo	
					RT + Bev + Nivo + Surgery	
	NCT03452579 [135,136]	2	Active Not Recruiting	rGBM	Nivo + Bev (10 mg/Kg)	OS-12: 41.1%, OS-12 (age > 60 year): 46.2%, OS-12 (age ≤ 60 years): 35.6%.
					Nivo + Bev (3 mg/Kg)	OS-12: 37.7%, OS-12 (age > 60 year): 23.8%, OS-12 (age ≤ 60 years): 56.4%.
	NCT04704154	2	Active Not Recruiting	Recurrent or Metastatic Tumors	Nivo + Regorafenib	
	NCT05909618	2	Not Yet Recruiting	GBM and Brain Metastases (MGMT-UN)	Crizanlizumab	
					Crizanlizumab + Nivo	
	NCT02617589 [137]	3	Completed	ndGBM (MGMT-UN)	Nivo + RT	mPFS: 6.0 months (95% CI, 5.7–6.2), mOS: 13.4 months (95% CI, 12.6–14.3)
					TMZ + RT	mPFS: 6.2 months (95% CI, 5.9–6.7), mOS: 14.9 months (95% CI, 13.3–16.1)
	NCT02667587 [138]	3	Active Not Recruiting	ndGBM (MGMT-M)	RT + TMZ + Nivo	mPFS: 10.64 months (95% CI, 8.90–11.79), mOS: 28.91 months (95% CI, 24.38–31.57),
					RT + TMZ + Placebo	mPFS: 10.32 months (95% CI, 9.69–12.45), mOS: 32.07 months (95% CI, 29.37–33.77),
Anti-PD-1 (Pembrolizumab)	NCT02852655	1	Completed	rGBM	Pembro	
	NCT02054806 [139]	1	Completed	Advanced Solid Tumors	Pembro	rGBM = mPFS: 2.8 months (95% CI, 1.9–8.1), mOS: 13.1 months (95% CI, 8.0–26.6)
	NCT05700955	1	Recruiting	rGBM	Pembro + TMZ	
	NCT02530502	1	Terminated	ndGBM	Pembro + TMZ + RT	
	NCT03722342 [140]	1	Active Not Recruiting	rGBM	Pembro + Olinvacimab	
	NCT03426891 [141]	1	Completed	ndGBM	Pembro + Vorinostat + TMZ + RT	
	NCT02311582 [142,143]	1/2	Active Not Recruiting	Recurrent Malignant Gliomas	Pembro + LITT	mPFS: 10.5 months, mOS: 11.4 months
					Pembro	mPFS: 2.1 months, mOS: 5.2 months
	NCT03277638 [144]	1/2	Recruiting	rGBM	Pembro (7 days before LITT)	
					Pembro (14 days after LITT)	
					Pembro (35 days after LITT)	
	NCT04977375	1/2	Recruiting	rGBM	Pembro + Stereotactic RT + Surgery	
	NCT02430363	1/2	Unknown	GBM or Gliosarcoma	Pembro	
					Pictilisib	
	NCT05053880	1/2	Unknown	rGBM	Pembro	
					Pembro + ACT001	
	NCT04121455 [145,146]	1/2	Active Not Recruiting	ndGBM (MGMT-UN)	NOX-A12 (200 mg) + RT	
					NOX-A12 (400 mg) + RT	
					NOX-A12 (600 mg) + RT	
					NOX-A12 (600 mg) + RT + Bev	
					NOX-A12 (600 mg) + RT	
					NOX-A12 (600 mg) + RT + Pembro	
	NCT05973903	1/2	Not Yet Recruiting	rGBM	Lenvatinib + Pembro + TTF	
	NCT02628067 [147]	2	Recruiting	Advanced Solid Tumors	Pembro	Glioma = mPFS: 1.4 (95% CI, 1.0–2.1), mOS: 5.6 months (95% CI, 2.6–16.2)
	NCT02337491 [148,149]	2	Completed	rGBM	Pembro + Bev	PFS-6: 26% (95% CI, 16.3–41.5), mOS: 8.8 months (95% CI, 7.7–14.2)
					Pembro	PFS-6: 6.7% (95% CI, 1.7–25.4), mOS: 10.3 months (95% CI, 8.5–12.5)
	NCT03661723 [150]	2	Active Not Recruiting	rGBM	Pembro + RT (lead-in)	ORR: 3.3%, OS-6: 83.3 (95% CI, 6.5–92.7), OS-12: 40.0 (95% CI, 22.8–56.6)
					Pembro + Bev + RT (lead-in)	ORR: 10.0%, OS-6: 56.7 (95% CI, 37.3–72.1), OS-12: 16.6 (95% CI, 6.0–31.7)
					Pembro + RT	
					Pembro + Bev + RT	
	NCT05463848	2	Recruiting	rGBM	Pembro + Olaparib + TMZ (Safety Lead In)	
					Pembro + Olaparib + TMZ (Surgical Cohort)	
					Pembro (Surgical Cohort)	
	NCT03347617	2	Active Not Recruiting	ndGBM	Ferumoxytol MRI + Pembro	
	NCT03197506	2	Suspended	ndGBM	Pembro + Surgery + TMZ + RT	
					Pembro + TMZ + RT	
	NCT05879120	2	Not Yet Recruiting	rGBM	MRgFUS + Neoadjuvant Pembro + Adjuvant Pembro	
					Neoadjuvant Pembro + Adjuvant Pembro	
	NCT03405792 [151]	2	Active Not Recruiting	ndGBM	TTF + TMZ + Pembro	mPFS: 12.0 months, PFS-12: 50.0%, mOS: 24.8 months, OS-24: 52.4%
					TTF + TMZ	mPFS: 5.8 months, PFS-12: 28.2%, mOS: 14.7 months, OS-24: 12%
	NCT02337686 [152]	2	Active Not Recruiting	rGBM	Pembro + Surgery	mPFS: 4.5 months (95% CI, 2.27–6.83), PFS-6: 40%, mOS: 20 months, estimated OS-12: 63%
	NCT05465954 [153]	2	Recruiting	rGBM	Pembro + Efineptakin alfa	
	NCT03797326 [154]	2	Active Not Recruiting	Solid Tumors	Pembro + Lenvatinib	
					Lenvatinib	
	NCT05235737	4	Recruiting	ndGBM	Neoadjuvant Pembro + Adjuvant Pembro + SOC	
					Neoadjuvant Pembro + SOC	
					SOC	
Anti-PD-L1 (Avelumab)	NCT03047473 [155]	2	Completed	ndGBM	Avelumab	ORR: 23.3%, mPFS: 9.7 months (95% CI, 8.2–15.5), mOS: 15.3 months (95% CI, 10.7–21.5)
Anti-PD-L1 (Atezolizumab)	NCT05423210	1	Active Not Recruiting	ndGBM	Atezo + Fractionated Stereotactic RT	
	NCT04160494	1	Active Not Recruiting	Recurrent Gliomas	D2C7-IT (6.92 μg/mL) + Atezo	
					D2C7-IT (4.61 μg/mL) + Atezo	
	NCT03158389	1/2	Completed	ndGBM (MGMT-UN)	APG101 + RT	
					Alectinib + RT	
					Idasanutlin + RT	
					Atezo + RT	
					Vismodegib + RT	
					Temsirolimus + RT	
					Palbociclib + RT	
	NCT03673787 [156]	1/2	Recruiting	Advanced Solid Tumors	Atezo + Ipatasertib	
	NCT03174197 [157]	1/2	Active Not Recruiting	ndGBM	Atezo + TMZ + RT	mOS: 17.1 months (95% CI, 13.9-N/A), mPFS: 9.7 months (95% CI, 7.6–15), mPFS (MGMT-M): 16.7 months (95% CI, 7.85-N/A), mPFS (MGMT-UN): 7.9 months (95% CI, 6.70–12.4)
	NCT05039281	1/2	Recruiting	rGBM	Atezo + Cabozantinib	
	NCT06069726	2	Not Yet Recruiting	rGBM	Pre-Surgery Atezo	
	NCT04729959	2	Suspended	rGBM	Atezo + Tocilizumab + Stereotactic RT	
					Atezo + Tocilizumab + Stereotactic RT + Surgery	
Anti-PD-L1 (Durvalumab)	NCT02336165 [158]	2	Completed	GBM	ndGBM = Durva + RT	OS-12: 60% (90% CI, 46.1–71.4)
					Bev-Naïve rGBM = Durva	PFS-6: 19.4% (90% CI, 9.3–32.1)
					Bev-Naïve rGBM = Durva + Bev (10 mg/Kg)	PFS-6: 15.2% (90% CI, 6.7–26.8)
					Bev-Naïve rGBM = Durva + Bev (3 mg/Kg)	PFS-6: 17.2% (90% CI, 7.7–29.7)
					Bev-Resistant rGBM = Durva + Bev	OS-6: 36.4% (80% CI, 23.5–49.3)
Anti-PD-1 + Anti-CTLA-4	NCT02311920 [159]	1	Completed	ndGBM or Gliosarcoma	TMZ + Ipi	
					TMZ + Nivo	
					TMZ + Ipi + Nivo	
	NCT04606316	1	Recruiting	rGBM	Nivo + Ipi	
					Nivo + Placebo	
					Placebo	
	NCT03233152 [160]	1	Active Not Recruiting	rGBM	Nivo + Ipi	mPFS: 11.7 weeks (2–152), mOS: 38 weeks (95% CI, 27–49),
	NCT06097975	1	Not Yet Recruiting	rGBM	Nivo + Ipi	
	NCT03367715	2	Completed	ndGBM (MGMT-UN)	Nivo + Ipi + Short-Course RT	OS-12: 80%, mOS: 16.85 months (4.5–32.9), mPFS: 5.92 months (1.5–13.9)
	NCT03430791	2	Terminated	rGBM	TTF + Nivo	
					TTF + Nivo + Ipi	
	NCT04817254	2	Recruiting	ndGBM	Nivo + Ipi (1 mg/Kg) + TMZ	
					Nivo + Ipi (3 mg/Kg) + TMZ	
	NCT04145115	2	Recruiting	rGBM	Nivo + Ipi	
	NCT04396860	2/3	Active, not recruiting	ndGBM (MGMT-UN)	RT + TMZ	
					RT + Nivo + Ipi	
	NCT02017717 [161,162]	3	Active, not recruiting	rGBM	Nivo	OS-12: 41.8% (95% CI, 34.7–48.8), mOS: 9.8 months (95% CI, 8.2–11.8), mPFS: 1.51 months (95% CI, 1.48–1.61)
					Nivo + Ipi	
					Bev	OS-12: 42.4% (95% CI, 34.9–49.6), mOS: 10.05 months (95% CI, 9–11.99), mPFS: 3.61 months (95% CI, 2.99–4.6)
Anti-PD-1 + Anti-GITR	NCT04225039 [163]	2	Active, not recruiting	rGBM	Retifanlimab + INCAGN01876 + Stereotactic RT	mPFS: 3.9 months (95% CI, 2.1–6.2), mOS: 9.4 months (95% CI, 8.2–10.6)
					Retifanlimab + INCAGN01876 + Stereotactic RT prior to Surgery	mPFS: 11.7 months, mOS: 20.1 months
					Retifanlimab + INCAGN01876 prior to Surgery	mPFS: 2.0 months, mOS: 9.4 months
Anti-PD-1 + Anti-IDO1	NCT04047706 [164]	1	Active, not recruiting	ndGBM	RT + TMZ + Nivo + BMS-986205	
					RT + Nivo + BMS-986205	
	NCT02327078 [165]	1/2	Completed	Advanced Tumors	Nivo + Epacadostat	
Anti-PD-1 + Anti-LAG-3	NCT03493932 [166]	1	Completed	GBM	Nivo + Relatlimab	
	NCT02658981 [167]	1	Completed	rGBM	BMS-986016	
					BMS-986016 + Nivo	
Anti-PD-1 + Anti-TIGIT	NCT04656535	0/1	Recruiting	GBM	Domvanalimab + Placebo	
					Zimberelimab + Placebo	
					Domvanalimab + Zimberelimab	
					Placebo	
	NCT04826393	1	Active Not Recruiting	Recurrent Gliomas	Cemiplimab + ASP8374	
Anti-PD-1 + Anti-TIM-3	NCT03961971	1	Active Not Recruiting	rGBM	Spartalizumab + Sabatolimab + Stereotactic RT	
Anti-PD-1 + Anti-GITR or Anti-IDO1 or Anti-CTLA-4	NCT03707457	1	Terminated	rGBM	Nivo + MK-4166	
					Nivo + Epacadostat	
					Nivo + Ipi	
Anti-PD-L1 + Anti-CTLA-4	NCT02794883	2	Completed	Recurrent Malignant Gliomas	Surgery + Durva	mOS: 11.71 (95% CI, 8.332–32.71), mPFS: 4.356 (95% CI, 2.941–32.74)
					Surgery + Tremelimumab	mOS: 7.246 (95% CI, 2.746–16.32), mPFS: 2.746 (95% CI, 2.68–8.727)
					Surgery + Durva + Tremelimumab	mOS: 7.703 (95% CI, 7.41–40.14), mPFS: 4.913 (95% CI, 2.905–120.4)
Various	NCT06047379	1/2	Not Yet Recruiting	Malignant Gliomas or Brain Metastases	NEO212 + Ipi	
					NEO212 + Pembro	
					NEO212 + Nivo	
					NEO212 + Regorafenib	
					NEO212 + CarbolaUn + Paclitaxel	
					NEO212 + FOLFIRI + Bev	
					NEO212	
					NEO212 + SOC	

Atezo, Atezolizumab; Bev, Bevacizumab; CI, confidence interval; CTLA-4, cytotoxic T-lymphocyte-associated protein 4; Durva, Durvalumab; GITR, glucocorticoid-induced TNFR-related protein; IDO1, Indoleamine 2,3-dioxygenase 1; IND, Indoximod; Ipi, Ipilimumab; LAG-3, Lymphocyte-Activation Gene 3; mOS, median overall survival; mPFS, median progression-free survival; MRgFUS, MRI-guided focused ultrasound; ndGBM, newly diagnosed GBM; Nivo, Nivolumab; ORR, objective response rate; OS-12, overall survival at 12 months; OS-24, overall survival at 24 months; Pembro, Pembrolizumab; PFS-6, progression-free survival at 6 months; PD-1, Programmed Cell Death-Protein 1; PD-L1, programmed Death-Ligand 1; rGBM, recurrent GBM; RT, radiotherapy; SOC, standard of care; TIGIT, T Cell immunoreceptor with Ig and ITIM domains; TIM-3, T cell immunoglobulin and mucin domain-containing protein 3; TMZ, Temozolomide; TTF, tumor-treating fields.

**Table 2 cancers-16-01276-t002:** List of clinical trials involving vaccination strategies in adult GBM patients. The table includes concluded or terminated studies, as well as those currently underway or preparing to enroll participants. Data were sourced from ClinicalTrials.gov, retrieved on 13 December 2023.

Antigen	Vaccine/ Delivery	NCT Number	Phase	Study Status	Tumor Target	Intervention	Outcome
CD133	DC vaccine	NCT02049489 [186]	1	Completed	rGBM	ICT-121	
CMV-pp65	Peptide Vaccine	NCT01854099	1	Withdrawn	ndGBM	TMZ (5 Day) + PEP-CMV (Day 6–8)	
						TMZ (5 Day) + PEP-CMV (day 22–24)	
						TMZ (21 Day) + PEP-CMV (day 22–24)	
	Peptide Vaccine	NCT02864368	1	Terminated	ndGBM	Td + TMZ (5 Day) + PEP-CMV (Component A + Component B) + Td	
						Td + TMZ (21 Day) + PEP-CMV (Component A + Component B) + Td	
						Td + TMZ (5 Day) + PEP-CMV (Safety Cohort) + Td	
						Td + TMZ (5 Day) + PEP-CMV (Component A) + Td	
						Td + TMZ (21 Day) + PEP-CMV (Component A) + Td	
	DC vaccine	NCT04963413	1	Active, not recruiting	ndGBM	CMV-DC + GM-CSF	
	DC vaccine	NCT00693095 [187]	1	Completed	ndGBM	CMV-ALT + CMV-DC	
						CMV-ALT	
	DC vaccine	NCT00626483 [188]	1	Completed	ndGBM	CMV-DC + GM-CSF + Basiliximab	mOS: 5.6 months (95% CI, 3.6–9.9), mPFS: 7.7 months (95% CI, 3.4–13.8)
	DC vaccine	NCT04741984	1	Withdrawn	ndGBM (MGMT-UN)	Monocyte loaded with mRNA encoding for CMV-pp65 (MT-201)	
	DC vaccine	NCT00639639 [189,190]	1	Completed	ndGBM	CMV-ALT + CMV-DC + Unpulsed DCs (or Td)	
						CMV-DC + Unpulsed DCs (or Td)	
						CMV-DC + GM-CSF + Unpulsed DCs (or Td)	
	DC vaccine	NCT02465268 [191]	2	Active, not recruiting	ndGBM	Td + TMZ + Short-Length CMV-DC + GM-CSF	
						Td + TMZ + Full-Length CMV-DC + GM-CSF	
						Unpulsed PBMCs	
	DC vaccine	NCT02366728 [192,193]	2	Completed	ndGBM	CMV-DC	mOS: 16 months (95% CI, 12.8–25.5), mPFS: 6.5 months (95% CI, 4.4–12.1)
						CMV-DC + Td	mOS: 20 months (95% CI, 16.7–25.6), mPFS: 6.7 months (95% CI, 4.6–15.2)
						CMV-DC + Td + Basiliximab	mOS: 19 months (95% CI, 10.2-N/A), mPFS: 7.1 months (95% CI, 6-N/A)
	Liposome	NCT04573140	1	Recruiting	ndGBM (MGMT-UN)	Liposome loaded with mRNA encoding for CMV-pp65 (RNA-LP)	
EGFRvIII	Peptide Vaccine	NCT00626015 [194]	1	Completed	ndGBM (EGFRvIII+)	Rindopepimut + TMZ + Daclizumab	
						Rindopepimut + TMZ + Placebo	
						Rindopepimut + Basiliximab	
	Peptide Vaccine	NCT01498328 [195]	2	Completed	rGBM (EGFRvIII+)	Bev-Naïve = Bev + Rindopepimut + GM-CSF	PFS-6: 28%, ORR: 30%, mDOR: 7.8 months (95% CI, 3.5–22.2)
						Bev-Naïve = Bev + KLH	PFS-6: 16%, ORR: 18%, mDOR: 5.6 months (95% CI, 3.7–7.4)
						Bev-Resistant = Bev + Rindopepimut + GM-CSF	
	Peptide Vaccine	NCT00458601 [196]	2	Completed	ndGBM (EGFRvIII+)	Rindopepimut + GM-CSF + TMZ	mOS: 21.8 months, OS-36: 26%
	Peptide Vaccine	NCT00643097 [197,198,199]	2	Completed	ndGBM (EGFRvIII+)	Rindopepimut + GM-CSF	mPFS: 14.2 (95% CI, 9.9–17.6)
						Rindopepimut + GM-CSF + TMZ (5 Day, 200 mg/m^2^)	mPFS: 12.1 (95% CI, 10.5–23.7)
						Rindopepimut + GM-CSF + TMZ (21 Day, 100 mg/m^2^)	mPFS: 11.6 (95% CI, 8.1–12.7)
	Peptide Vaccine	NCT01480479 [200]	3	Completed	ndGBM (EGFRvIII+)	Rindopepimut + GM-CSF + TMZ	mOS: 20.1 months (95% CI, 18.5–22.1)
						KLH + TMZ	mOS: 20.0 months (95% CI, 18.1–21.9)
HSPPC-96	Peptide Vaccine	NCT00293423 [201,202]	1/2	Completed	Recurrent Gliomas	HSPPC-96 Vaccine	OS-12: 29.3% (95% CI, 16.6–45.7), mOS: 42.6 weeks (95% CI, 34.7–50.5)
	Peptide Vaccine	NCT00905060 [203]	2	Completed	ndGBM	HSPPC-96 Vaccine + TMZ	mOS: 23.8 months (95% CI, 9.8–30.2), mPFS: 18 (95% CI, 12.4–21.8)
	Peptide Vaccine	NCT01814813 [204]	2	Terminated	rGBM	HSPPC-96 Vaccine + Concomitant Bev	mOS: 6.6 months (95% CI, 5.4–10.4), mPFS: 3.7 months (95% CI, 2.9–5.4)
						HSPPC-96 Vaccine + Bev At Progression	mOS: 9.2 months (95% CI, 5.7–11.6), mPFS: 2.5 months (95% CI, 2.0–3.5)
						Bev	mOS: 10.7 months (95% CI, 8.8–17.2), mPFS: 5.3 months (95% CI, 3.7–8.0)
hTERT	Peptide Vaccine	NCT00069940	1	Completed	Sarcoma and Brain Tumors (HLA-A2+)	540–548 hTERT Vaccine + GM-CSF	
	Peptide Vaccine	NCT04280848 [205]	2	Active, not recruiting	ndGBM (MGMT-UN)	MGMT-UN = UCPVax	mPFS: 8.9 months (95% CI, 7.6–10.6), mOS: 17.9 months (95% CI, 16–23), OS-24: 26%
						MGMT-UN or MGMT m = UCPVax + TMZ	
Survivin	Peptide Vaccine	NCT01250470 [206]	1	Completed	Recurrent Malignant Gliomas	SurVaxM/Montanide ISA-51 + GM-CSF	mPFS: 17.6 weeks, mOS: 86.6 weeks
	Peptide Vaccine	NCT05163080 [207]	2	Recruiting	ndGBM	SurVaxM/Montanide ISA-51 + GM-CSF + TMZ	
						Placebo/Montanide ISA-51 + GM-CSF + TMZ	
	Peptide Vaccine	NCT02455557 [208]	2	Active, not recruiting	ndGBM	SurVaxM/Montanide ISA-51 + GM-CSF + TMZ	PFS-6: 95% (95% CI, 86–98), mPFS: 11.4 months, mOS: 25.8 months (95% CI, 19.5–43.5)
AIM-2, MAGE-1, HER2/neu, TRP-2, gp100, and IL-13Rα2	DC vaccine	NCT01280552 [209]	2	Completed	ndGBM	ICT-107	mOS: 18.3 months (95% CI, 14.9–21.2), mPFS: 11.2 months (95% CI, 8.2–13.0)
						Unpulsed DCs	mOS: 16.7 months (95% CI, 12.3–23.0), mPFS: 9.0 months (95% CI, 5.5–10.3)
		NCT02546102	3	Suspended	ndGBM	ICT-107 + TMZ	
						Placebo + TMZ	
EGFRvIII, IL-13Rα2, EphA2, HER2/neu, YKL-40	Peptide Vaccine	NCT02754362	2	Withdrawn	rGBM	Bev + Multipeptide Vaccine + Poly-ICLC	
EphA2, CMV-pp65, and Survivin	Peptide Vaccine	NCT05283109	1	Recruiting	ndGBM (MGMT-UN)	P30-EPS + Poly-ICLC	
BCAN, CSPG4, FABP7, IGF2BP3, NLGN4X, NRCAM, PTPRZ1 (2 peptides), and TNC	Peptide Vaccine	NCT01403285	1	Terminated	GBM (HLA-A2+)	IMA950 + GM-CSF + Imiquimod + Cyclophosphamide	
	Peptide Vaccine	NCT01222221 [210]	1	Completed	ndGBM (HLA-A2+)	IMA950 + GM-CSF + Chemoradiotherapy (Vaccine before TMZ)	mOS: 14.4 months
						IMA950 + GM-CSF + Chemoradiotherapy (Vaccine after TMZ)	mOS: 15.7 months
	Peptide Vaccine	NCT01920191 [211,212]	1/2	Completed	ndGBM (HLA-A2+)	IMA950 + Poly-ICLC	mOS: 19 months (95% CI: 17.25–27.87), PFS-6: 81%, mPFS: 9.5 months
WT-1, PSMA, hTERT, IL-2	Electroporation	NCT03491683 [213]	1/2	Active, not recruiting	ndGBM	MGMT-UN = INO-5401 + INO-9012 + Cemiplimab + RT + TMZ	mOS: 17.9 months (95% CI, 14.5–19.8)
						MGMT m = INO-5401 + INO-9012 + Cemiplimab + RT + TMZ	mOS: 32.5 months (95% CI, 18.4-N/A)
Tumor Lysate	DC vaccine	NCT01171469 [214]	1	Completed	Recurrent or Progressive Malignant Gliomas	DCs pulsed with Tumor Lysate (from BTSCs) + Imiquimod	
	DC vaccine	NCT00068510 [215]	1	Completed	Malignant Gliomas	DCs pulsed with Tumor Lysate	
	DC vaccine	NCT01808820	1	Completed	Malignant Gliomas	DCs pulsed with Tumor Lysate + Imiquimod	
	DC vaccine	NCT02010606 [216]	1	Completed	GBM	ndGBM = DCs pulsed with Tumor Lysate (from Allogeneic Stem-like Cells) + RT + TMZ	mPFS: 8.75 months, mOS: 20.36 months
						rGBM = DCs pulsed with Tumor Lysate (from Allogeneic Stem-like Cells) + Bev (optional)	mPFS: 3.23 months, PFS-6: 24%, mOS: 11.97 months
	DC vaccine	NCT01213407 [217]	2	Completed	Malignant Gliomas	SOC + DCs pulsed with Tumor Lysate (Trivax)	
						SOC	
	DC vaccine	NCT01006044 [218]	2	Completed	GBM	DCs pulsed with Tumor Lysate	mPFS: 12.7 months (95% CI, 7–16), mOS: 23.4 months (95% CI, 16–33.1)
	DC vaccine	NCT00323115 [219]	2	Completed	ndGBM	DCs pulsed with Tumor Lysate + RT + TMZ	PFS-6: 90%, mPFS: 9.5 months, mOS: 28 months
	DC vaccine	NCT00045968 [220,221]	3	Active, not recruiting	GBM	DCs pulsed with Tumor Lysate (DCVax-L)	ndGBM = mOS: 19.3 months (95% CI, 17.5–21.3) rGBM = mOS: 13.2 months (95% CI, 9.7–16.8)
						Unpulsed PBMCs	ndGBM = mOS: 16.5 months (95% CI, 16.0–17.5) rGBM = mOS: 7.8 months (95% CI, 7.2–8.2)
Personalized	Peptide Vaccine	NCT02149225 [222,223]	1	Completed	ndGBM	APVAC1/APVAC2 + Poly-ICLC + GM-CSF + TMZ	mPFS: 14.2 months, mOS: 29 months
	Peptide Vaccine	NCT02510950	1	Terminated	ndGBM	Personalized Peptide Vaccine + Poly-ICLC + TMZ	
	Peptide Vaccine	NCT03223103 [224]	1	Active, not recruiting	ndGBM	Mutation-derived Tumor Antigen Vaccine + Poly-ICLC + TTF	Estimated PFS-12: 62.5%, estimated OS-12: 83.3%
	Peptide Vaccine	NCT05557240	1	Recruiting	ndGBM	NPVAC1 + Poly-ICLC + TMZ	
						NPVAC2 + Poly-ICLC + TMZ	
	Electroporation	NCT04015700	1	Active, not recruiting	ndGBM (MGMT-UN)	Personalized DNA Vaccine (GNOS-PV01) + INO-9012	
	Peptide Vaccine	[225]	3	Concluded	rGBM (HLA-A24+)	Personalized Peptide Vaccine	mOS: 8.4 months (95% CI, 6.6–10.6)
						Placebo	mOS: 8.0 months (95% CI, 4.8–12.9)
N/A	Peptide Vaccine	NCT04842513	1	Recruiting	ndGBM (HLA-A2+, MGMT-M)	Multipeptide Vaccine + XS15 + Montanide ISA-51	
	DC vaccine	NCT04968366	1	Recruiting	ndGBM	DCs pulsed with Multiple Neopeptides + TMZ	
	DC vaccine	NCT00612001 [215]	1	Completed	Malignant Gliomas	DCs pulsed with Multiple Glioma-associated Peptides	
	DC vaccine	NCT00890032 [226]	1	Completed	rGBM	DCs pulsed with mRNA from BTSCs	mPFS: 3.2 months (95.0% CI, 1.8–7.2), mOS: 11 months (95.0% CI, 8.2–14.8)
	DC vaccine	NCT02820584	1	Completed	rGBM	DCs pulsed with mRNA from Glioma Stem Cells	
	DC vaccine	NCT00846456 [227]	1/2	Completed	GBM	DCs pulsed with mRNA from Glioma Stem Cells	mOS (treated group): 759 days, mOS (control group): 585 days
	DC vaccine	NCT00576641 [228]	1	Completed	Brain Stem Glioma and GBM	DCs pulsed with Tumor Peptides	

Bev, Bevacizumab; BTSC, brain tumor stem cell; CAR-T, chimeric antigen receptor T cell; CI, confidence interval; CMV-ALT, CMV-autologous lymphocyte transfer; DC, dendritic cell; GM-CSF, granulocyte-macrophage colony-stimulating factor; KLH, Keyhole Limpet Haemocyanin; mDOR, median duration of response; mOS, median overall survival; mPFS, median progression-free survival; ndGBM, newly diagnosed GBM; NPVAC, NeoPep vaccine; ORR, objective response rate; OS-12, overall survival at 12 months; OS-24, overall survival at 24 months; PBMC, peripheral blood mononuclear cell; Poly-ICLC, polyinosinic–polycytidylic acid stabilized with polylysine and carboxymethylcellulose; PFS-12, progression-free survival at 12 months; PFS-6, progression-free survival at 6 months; rGBM, recurrent GBM; RT, radiotherapy; SOC, standard of care; Td, tetanus and diphtheria toxoid; TMZ, Temozolomide.

**Table 3 cancers-16-01276-t003:** List of clinical trials involving adoptive T cell therapies in adult GBM patients. The table includes concluded or terminated studies, as well as those currently underway or preparing to enroll participants. Data were sourced from ClinicalTrials.gov, retrieved on 13 December 2023.

Antigen		NCT Number	Phase	Study Status	Tumor Target	Intervention	Outcome
Monovalent CAR-T	B7-H3	NCT05241392	1	Recruiting	rGBM	B7-H3 CAR-T	
		NCT05366179	1	Recruiting	rGBM	B7-H3 CAR-T	
		NCT05474378	1	Recruiting	rGBM	B7-H3 CAR-T	
		NCT04385173	1	Recruiting	rGBM or Refractory GBM	B7-H3 CAR-T + TMZ	
		NCT04077866	1/2	Recruiting	rGBM or Refractory GBM	TMZ	
						TMZ + B7-H3 CAR-T	
	CD70	NCT05353530	1	Recruiting	ndGBM (MGMT-UN, CD70+)	CD70 CAR-T	
	EGFRvIII	NCT05802693	1	Not yet recruiting	rGBM (EGFRvIII+)	EGFRvIII CAR-T	
		NCT02209376 [253,254,255]	1	Terminated	rGBM	EGFRvIII CAR-T	mOS: 251 days
		NCT02664363 [256]	1	Terminated	ndGBM (EGFRvIII+)	EGFRvIII CAR-T	
		NCT02844062	1	Unknown	rGBM (EGFRvIII+)	EGFRvIII CAR-T	
		NCT03283631	1	Terminated	rGBM	EGFRvIII CAR-T	
		NCT05063682	1	Unknown	Leptomeningeal GBM (EGFRvIII+)	EGFRvIII CAR-T	
		NCT05660369	1	Recruiting	GBM	EGFRvIII BiTE-secreting CAR-T	
		NCT05024175	Observational	Not yet recruiting	GBM	/	
		NCT01454596 [257]	1/2	Completed	Malignant Gliomas (EGFRvIII+)	EGFRvIII CAR-T	mOS: 6.9 months (2.8–10)
		NCT03941626	1/2	Unknown	Solid Tumors (EGFRvIII+)	EGFRvIII CAR-T	
		NCT03638206	1/2	Unknown	Solid Tumors (EGFRvIII+)	EGFRvIII CAR-T	
	EMMPRIN	NCT04045847	1	Unknown	Recurrent Malignant Gliomas (CD147+)	EMMPRIN CAR-T	
	GD2	NCT03170141 [258]	1	Enrolling by invitation	rGBM (GD2+)	GD2 CAR-T	mOS = 10 months (3–24)
	HER2/neu	NCT01109095 [259]	1	Completed	GBM	HER2 CAR-T	
		NCT03389230	1	Active, not recruiting	Recurrent or Refractory Malignant Gliomas	HER2 CAR-T	
	IL-13Rα2	NCT02208362 [260]	1	Active, not Recruiting	Recurrent Malignant Gliomas	IL-13Rα2 CAR-T (intratumoral)	
						IL-13Rα2 CAR-T (intracavitary)	
						IL-13Rα2 CAR-T (intraventricular)	
						IL-13Rα2 CAR-T (intratumoral/intraventricular)	
		NCT04661384	1	Recruiting	Leptomeningeal GBM, Ependymoma, or Medulloblastoma	IL-13Rα2 CAR-T	
		NCT05540873	1	Recruiting	Recurrent or Refractory Malignant Gliomas	IL-13Rα2 CAR-T	
		NCT00730613 [261]	1	Completed	Recurrent or Refractory Malignant Gliomas	IL-13Rα2 CTLs	
	MMP2 (Chlorotoxin)	NCT04214392	1	Recruiting	rGBM (MMP2+)	MMP2 CAR-T (intratumoral)	
						MMP2 CAR-T (intratumoral/intraventricular)	
		NCT05627323 [262]	1	Recruiting	rGBM (MMP2+)	MMP2 CAR-T	
	NKG2D	NCT04270461	1	Withdrawn	Recurrent Solid Tumors (NKG2DL+)	NKG2D CAR-T	
		NCT05131763	1	Recruiting	Recurrent Solid Tumors (NKG2DL+)	NKG2D CAR-T	
		NCT04717999	N/A	Not yet recruiting	rGBM	NKG2D CAR-T	
		NCT04550663	1	Unknown	Relapsed or Refractory Solid Tumors (NKG2DL+)	NKG2D CAR-T	
	PD-L1	NCT02937844	1	Unknown	rGBM	PD-L1 CAR-T	
	SNC-109	NCT05868083	1	Recruiting	rGBM	SNC-109 CAR-T	
Polyvalent CAR-T	IL-7Ra, CD44 and CD133	NCT05577091	1	Not yet recruiting	rGBM	Tris-CAR-T	
	EGFRvIII, IL-13Rα2, HER2/neu, EphA2, CD133, GD2	NCT03423992 [263]	1	Unknown	Recurrent Malignant Gliomas	Personalized CAR-T	mOS (EphA2-specific CAR-T) = 86–181 days
TILs		NCT05333588	1	Recruiting	GBM	TILs	
		NCT03347097 [264]	1	Unknown	rGBM	TILs	mOS: 16.1 months
						PD-1-TILs	mOS: 11.2 months
		NCT04943913	1	Recruiting	Gliomas	TILs	

BiTE, bispecific T-cell engager; CAR-T, chimeric antigen receptor T cell; MGMT-unmethylated, MGMT-UN; mOS, median overall survival; ndGBM, newly diagnosed GBM; rGBM, recurrent GBM; TIL, tumor-infiltrating lymphocyte.

**Table 4 cancers-16-01276-t004:** List of clinical trials involving virus-based therapies in adult GBM patients. The table includes concluded or terminated studies, as well as those currently underway or preparing to enroll participants. Data were sourced from ClinicalTrials.gov, retrieved on 13 December 2023.

Virus Name	Virus Type		NCT Number	Phase	Study Status	Tumor Target	Intervention	Outcome
Adenovirus	OV	CRAd-S-pk7	NCT05139056	1	Recruiting	Recurrent Malignant Gliomas	NSC-expressing CRAd-S-pk7	
			NCT03072134 [281]	1	Completed	Newly Diagnosed Malignant Gliomas	NSC-expressing CRAd-S-pk7	mPFS: 9.1 months (95% CI, 8.5–36), mOS: 18.4 months (95% CI, 6.5–36)
		DNX-2401	NCT03896568 [282]	1	Recruiting	Recurrent Malignant Gliomas	BM-hMSCs loaded with DNX-2401	
			NCT02197169 [283]	1	Completed	rGBM or Gliosarcoma	DNX-2401	
							DNX-2401 + IFN-γ	
			NCT01956734 [284]	1	Completed	rGBM	DNX-2401 + TMZ	
			NCT01582516	1/2	Completed	rGBM	DNX-2401	
			NCT00805376 [283]	1	Completed	Recurrent Malignant Gliomas	DNX-2401	mOS: 9.5 months
							DNX-2401 + Surgery	mOS: 13 months
		DNX-2440	NCT03714334	1	Terminated	rGBM	DNX-2440	
		ONYX-015	[285]	1	Completed	Recurrent Malignant Gliomas	ONYX-015	mOS (all patients): 6.2 months (1.3–28.0), mOS (GBM patients): 4.9 months
	Non-Lytic	AdV-ECRT-122T	NCT06102525	1/2	Not yet recruiting	GBM (hTERT+)	AdV-ECRT-122T + Valganciclovir	
		AdV-HSV-TK	NCT00002824	1	Completed	Primary Brain Tumors	AdV-HSV-TK + Ganciclovir	
			NCT01811992 [286]	1	Completed	Malignant Gliomas	AdV-HSV-TK + AdV-Flt3L + Valacyclovir	mOS: 21.3 months (95% CI, 11.1–26.1)
			NCT00751270 [287]	1	Completed	Malignant Gliomas	Resectable Gliomas = AdV-HSV-TK + Valacyclovir + RT	
							Unresectable Gliomas = AdV-HSV-TK + Valacyclovir + RT	
			NCT03596086	1/2	Recruiting	rGBM	AdV-HSV-TK + Valacyclovir + Radiochemotherapy	
			NCT03603405	1/2	Recruiting	ndGBM	AdV-HSV-TK + Valacyclovir + Radiochemotherapy	
			NCT00870181 [288]	2	Completed	Recurrent Malignant Gliomas	AdV-HSV-TK + Ganciclovir	PFS-6: 71.4%, mPFS: 34.9 weeks (9.0–238.4), mOS: 45.7 weeks (9.0–238.4)
							SOC	PFS-6: 5.6%, mPFS: 7.4 weeks (1.1–35.3), mOS: 8.6 weeks (1.1–45.0)
			NCT00589875 [289]	2	Completed	Malignant Gliomas	AdV-HSV-TK + Valacyclovir + RT	mOS: 17.1 months
							SOC	mOS: 13.5 months
		Ad-RTS-IL-12	NCT02026271 [290]	1	Completed	Malignant Gliomas	Ad-RTS-IL-12 + Veledimex	
		AdV-IFN-β	NCT05914935	1	Recruiting	rGBM	AdV-IFN-β	
			NCT00031083	1	Completed	Malignant Gliomas	AdV-IFN-β	
		AdV-p53	NCT00004041	1	Completed	Recurrent Malignant Gliomas	AdV-p53	
			NCT00004080	1	Completed	Recurrent or Progressive Brain Tumors	AdV-p53	
		VB-111	NCT01260506 [291]	1/2	Completed	rGBM	VB-111 until progression	mOS: 223 days, OS-12: 18%
							VB-111 upon progression + Bev (primed combination)	mOS: 414 days, OS-12: 57%
							VB-111 + Bev (unprimed combination)	mOS: 141.5 days, OS-12: 10%
			NCT02511405 [292]	3	Completed	rGBM	VB-111 + Bev	mOS: 6.8 months, ORR: 27.3%
							Bev	mOS: 7.9 months, ORR: 21.9%
Herpes Simplex Virus	OV	C134	NCT03657576	1	Recruiting	rGBM	C134	
		C5252	NCT05095441	1	Not yet recruiting	rGBM or Progressive GBM	C5252	
		HSV-1716	NCT02031965	1	Terminated	Recurrent Malignant Gliomas	HSV-1716	
			[293]	1	Completed	Recurrent Malignant Gliomas	HSV-1716	
			[294]	1	Completed	Malignant Gliomas	HSV-1716	
			[295]	1	Completed	Malignant Gliomas	HSV-1716	
		G207	NCT00157703 [296]	1	Completed	Malignant Gliomas	G207 + RT	mOS: 7.5 months (95% CI, 3.0–12.7)
			NCT00028158 [297]	1/2	Completed	Recurrent Brain Tumors	G207	
			NCT00036699 [298]	1/2	Completed	Recurrent Brain Tumors	G207	
		G47Δ	UMIN000002661 [299]	1/2	Completed	rGBM or Progressive GBM	G47Δ	mOS: 30.5 (95% CI, 19.2–52.7)
		M032	NCT02062827	1	Active, not recruiting	Recurrent Malignant Gliomas	M032 (NSC 733972)	
		rQnestin34.5v.2	NCT03152318 [300,301]	1	Recruiting	Recurrent Malignant Gliomas	rQNestin34.5v.2	
							rQNestin34.5v.2 + Cyclophosphamide	
							rQNestin34.5v.2 (Multiple doses)	
Retrovirus	Non-Lytic	RV-HSV-TK	[302]	3	Completed	ndGBM	SOC	mOS: 354 days (95% CI, 315–372), OS-12: 55%
							SOC + RV-HSV-TK + Ganciclovir	mOS: 365 days (95% CI, 334–416), OS-12: 50%
		Toca 511	NCT01985256 [303]	1	Completed	Recurrent Brain Tumors	Toca 511 + 5-FC	
			NCT02576665 [304]	1	Terminated	Solid Tumors or Lymphomas	Toca 511 + 5-FC	
			NCT01470794 [305,306]	1	Completed	Recurrent Malignant Brain Tumors	Toca 511 + 5-FC	
			NCT01156584 [307]	1	Completed	Recurrent Malignant Gliomas	Toca 511 + 5-FC	
			NCT04327011	1	Terminated	/	Toca 511 + 5-FC (Long term safety follow-up)	
			NCT02414165 [308]	2/3	Terminated	Recurrent Malignant Gliomas	Toca 511 + 5-FC	mOS: 11.10 months
							Lomustine, TMZ or Bev	mOS: 12.22 months
			NCT04105374 [309]	2/3	Withdrawn	ndGBM	SOC	
							SOC + Toca 511 + 5-FC	
Measles Virus	OV	MV-CEA	NCT00390299	1	Completed	rGBM	MV-CEA (Intracavitary)	PFS-6: 22.2% (95% CI, 6.6–75.4), mOS: 11.8 months (95% CI, 4.4-N/A)
							MV CEA (Intratumoral/Intracavitary)	PFS-6: 23.1% (95% CI, 8.6–62.3), mOS: 11.4 months (95% CI, 4.3-N/A)
Newcastle Disease Virus	OV	NDV-HUJ	NCT01174537 [310]	1/2	Withdrawn	rGBM, Sarcoma or Neuroblastoma	NDV (HUJ strain)	
		NDV-MTH-68/H	[311]	/	/	Malignant Gliomas	NDV (MTH-68/H strain)	
Parvovirus	OV	H-1PV	NCT01301430 [312,313]	1/2	Completed	rGBM or Progressive GBM	H-1PV	
Poliovirus	OV	PVSRIPO	NCT01491893 [314]	1	Completed	rGBM	PVSRIPO	mOS (PVSRIPO): 12.5 months (95% CI, 9.9–15.2), mOS (historical controls): 11.3 months (95% CI, 9.8–12.5)
			NCT02986178	2	Active, not recruiting	Recurrent Malignant Gliomas	PVSRIPO	
Reovirus	OV	Reolysin	NCT00528684	1	Completed	Malignant Gliomas	Reolysin	
			[315]	1	Completed	Recurrent Malignant Gliomas	Reolysin	mOS: 21 weeks (6 to 234)
			[316]	1	Completed	Recurrent Malignant Gliomas	Reolysin	mOS: 140 days (97 to 989)
			[317]	1	Completed	Malignant Gliomas and Brain Metastases	Reolysin	mOS: 469 days (118 to 1079)
Vaccinia Virus	OV	TG6002	NCT03294486	1/2	Completed	rGBM	TG6002 + 5-FC	

5-FC, 5-FluoroCytosine; AdV, Adenovirus; Bev, Bevacizumab; BM-hMSCs, allogeneic bone marrow-derived human mesenchymal stem cells; CI, confidence interval; HSV, herpes simplex virus; MGMT-methylated, MGMT-M; MGMT-unmethylated, MGMT-UN; mOS, median overall survival; OV, oncolytic virus; mPFS, median progression-free survival; MV, measles virus; ndGBM, newly diagnosed GBM; NDV, Newcastle disease virus; NSC, neural stem cells; ORR, objective response rate; OS-12, overall survival at 12 months; PFS-6, progression-free survival at 6 months; rGBM, recurrent GBM; RT, radiotherapy; RV, retrovirus; SOC, standard of care; TMZ, Temozolomide.

**Table 5 cancers-16-01276-t005:** List of clinical trials combining immunotherapeutic strategies in adult GBM patients. The table includes concluded or terminated studies, as well as those currently underway or preparing to enroll participants. Data were sourced from ClinicalTrials.gov, retrieved on 13 December 2023.

Combination		NCT Number	Phase	Study Status	Tumor Target	Intervention	Outcome
ICT + ACT	Anti-PD-1 + CAR-T	NCT03726515	1	Completed	ndGBM (MGMT-UN)	EGFRvIII CAR-T + Pembro	
		NCT04003649	1	Recruiting	rGBM or Refractory GBM	Nivo + IL-13Rα2 CAR-T + Ipi	
						Nivo + IL-13Rα2 CAR-T	
						IL-13Rα2 CAR-T	
ICT + Vaccine	Anti-PD-1 + CMV-DC	NCT02529072	1	Completed	Recurrent Brain Tumors	Nivo + Surgery + Nivo&CMV-DC	
						Nivo&CMV-DC + Surgery + Nivo&CMV-DC	
	Anti-PD-1 + HSPPC-96	NCT03018288	2	Completed	ndGBM (MGMT-UN)	RT + TMZ	
						RT + TMZ + Pembro	
						RT + TMZ + Pembro + HSPPC-96 Vaccine	
						RT + TMZ + Pembro + Placebo	
	Anti-PD-1 + IMA950	NCT03665545 [364]	1/2	Active, not recruiting	rGBM	IMA950 + Poly-ICLC	
						IMA950 + Poly-ICLC + Pembro	
	Anti-PD-1 or Anti-CTLA-4 + NeoVax	NCT03422094	1	Terminated	ndGBM (MGMT-UN)	NeoVax + Nivo (start at time of progression)	
						NeoVax + Nivo (start with Cycle 1)	
						NeoVax + Nivo (start with Cycle 2)	
						NeoVax + Ipi + Nivo (start with Cycle 3)	
						NeoVax + Ipi + Nivo (day 1&15 each cycle)	
		NCT02287428 [237,365]	1	Recruiting	ndGBM	RT + NeoVax	mPFS: 7.6 months (90% CI, 6.2–9.5), mOS: 16.8 months (90% CI, 9.6–21.3)
						RT + Pembro followed by NeoVax + Pembro	
						RT followed by NeoVax + Pembro	
						RT + 1 dose Pembro followed by NeoVax + Pembro	
						MGMT m = RT + TMZ Followed by TMZ + NeoVax + Pembro	
	Anti-PD-1 + SurVaxM	NCT04013672 [366]	Phase 2	Active, not recruiting	rGBM	Pembro + SurVaxM/Montanide ISA-51 + GM-CSF (no prior immunotherapy)	
						Pembro + SurVaxM/Montanide ISA-51 + GM-CSF (prior failed immunotherapy)	
	Anti-PD-1 + DC-Tumor Lysate	NCT03014804	2	Withdrawn	rGBM	DCVax-L	
						DCVax-L + Nivo	
		NCT04201873	1	Recruiting	rGBM	Pembro + ATL-DC + Poly-ICLC	
						Placebo + ATL-DC + Poly-ICLC	
ICT + Virus	Anti-PD-1 + AdV	NCT03576612	1	Active, not recruiting	Newly Diagnosed Malignant Gliomas	MGMT-UN = AdV-HSV-TK/Valacyclovir + RT + TMZ + Nivo	
						MGMT m and undetermined = AdV-HSV-TK/Valacyclovir + RT + TMZ + Nivo	
		NCT03636477 [367]	1	Completed	rGBM or Progressive GBM	Ad-RTS-IL-12 + Veledimex + Nivo	mOS: 16.9 months
						Nivo	mOS: 9.8 months
	Anti-PD-1 + HSV	NCT05084430	1/2	Recruiting	Recurrent Malignant Gliomas	rGBM = Pembro + M032	
						ndGBM = Pembro + M032	
		NCT02798406 [337]	2	Completed	rGBM or Gliosarcoma	DNX-2401 + Pembro	ORR: 10.4% (90% CI, 4.2–20.7), OS-12: 52.7% (95% CI, 40.1–69.2), mOS: 12.5 months (10.7–13.5)
	Anti-PD-1 + Poliovirus	NCT04479241 [368]	2	Active, not recruiting	rGBM	PVSRIPO + Pembro	
	Anti-PD-L1 + Poliovirus	NCT03973879	1/2	Withdrawn	Recurrent Malignant Gliomas	PVSRIPO + Atezo	

ACT, adoptive cell therapy; AdV, Adenovirus; Atezo, Atezolizumab; CAR-T, chimeric antigen receptor T cells; CI, confidence interval; DC, dendritic cell; HSV, herpes simplex virus; ICT, immune checkpoint therapy; Ipi, Ipilimumab; MGMT-methylated, MGMT-M; MGMT-unmethylated, MGMT-UN; mOS, median overall survival; mPFS, median progression-free survival; ndGBM, newly diagnosed GBM; ORR, objective response rate; Nivo, Nivolumab; OS-12, overall survival at 12 months; Pembro, Pembrolizumab; rGBM, recurrent GBM; RT, radiotherapy; TMZ, temozolomide.

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
