# Peer review of "Immunotherapeutic Strategies for the Treatment of Glioblastoma: Current Challenges and Future Perspectives"

_cancers, 2024, doi:10.3390/cancers16071276_

Round 1
Reviewer 1 Report
Comments and Suggestions for Authors This review is well structured, detailed and balanced. I have a few comments that could improve the discussion of certain issues: 1) Lines 578-581. The design of phase III clinical trial of the vaccine DCVax-L was significantly criticized. Please consider the following comments and reviewsdoi: 10.3390/cancers15123251
doi: 10.1016/j.neurol.2023.03.014
doi: 10.1038/s41416-023-02194-1
doi: 10.1093/neuonc/noac281
doi: 10.21037/tcr-23-603
doi: 10.1001/jamaoncol.2023.1069
doi: 10.1001/jamaoncol.2023.1066
2) Line 742. “ONYX-015 contains a deletion of the E1B gene that restricts virus replication to p53-negative tumor cells”. The original hypothesis that E1b55K-deleted oncolytic Ads could only replicate in p53-deficient tumor cells but not in normal cells with functional p53 was revisited/disproved (please see DOI: 10.1016/j.ccr.2004.11.012 and a detailed review doi: 10.3390/v7112905).
3) Lines 747-749. “Of note, ~50% of gliomas maintain p53 expression therefore limiting the effect of the treatment to half of the tumor cases”. This statement in the context of ONYX-015 has not been proved (see above the comment #2). Remove it.
4) Lines 753-754 “These modifications make DNX-2401 able to target and replicate within glioma cells lacking Rb expression [324, 325].” The original hypothesis that in normal nonproliferating cells with intact pRb, replication of Ad5 with the delta-24 modification is significantly attenuated (DOI: 10.1038/sj.onc.1203251; DOI: 10.1038/80474) has not not been confirmed by many researchers (please see the introduction and discussion sections in doi: 10.1016/j.omto.2021.12.013)
5) In light of the fact that for a number of viruses the initial hypotheses on the mechanisms of selectivity towards tumor cells were not confirmed later, I would recommend that the authors more carefully study the history of each virus. It is possible that other unproven/contradictory/debatable statements might be identified.
6) Standard radiochemotherapy-promoted systemic immunosuppression (iatrogenic) is a significant barrier to immunotherapies. Although different immunosuppressive mechanisms operate in patients with glioblastoma/gliomas at presentation, clinical data provide compelling evidence that the immunological competence of patients may be significantly compromised by standard therapy, exacerbating tumor-related systemic immunosuppression. Standard therapy (radiation, temozolomide, steroids) affects diverse immune cell subsets, with primary immunodeficiency related to long-lasting T cell lymphopenia. Changes in the standard therapy (the need for shifting the treatment paradigm) are required to increase the success of immunotherapies (e.g., discussed in doi: 10.3389/fimmu.2024.1326757). In the section Conclusions and Future Directions, the authors could briefly discuss this issue.
Author Response
Response to Reviewer #1:
This review is well structured, detailed and balanced.
Authors: Thank you for your positive feedback.
I have a few comments that could improve the discussion of certain issues:
1) Lines 578-581. The design of phase III clinical trial of the vaccine DCVax-L was significantly criticized. Please consider the following comments and reviews
doi: 10.3390/cancers15123251
doi: 10.1016/j.neurol.2023.03.014
doi: 10.1038/s41416-023-02194-1
doi: 10.1093/neuonc/noac281
doi: 10.21037/tcr-23-603
doi: 10.1001/jamaoncol.2023.1069
doi: 10.1001/jamaoncol.2023.1066
Authors: We would like to thank the reviewer for this comment. We have revised the text emphasizing the concerns that the DCVax-L trial raised among the scientific community (Lines 575-582 and 587-592). We also added the suggested publications.
2) Line 742. “ONYX-015 contains a deletion of the E1B gene that restricts virus replication to p53-negative tumor cells”. The original hypothesis that E1b55K-deleted oncolytic Ads could only replicate in p53-deficient tumor cells but not in normal cells with functional p53 was revisited/disproved (please see DOI: 10.1016/j.ccr.2004.11.012 and a detailed review doi: 10.3390/v7112905).
Authors: We thank the reviewer for raising this point, and we have revised the text accordingly, saying that ONYX-015 preferentially replicates in cancer cells through various not yet fully characterized mechanisms. We have added the suggested publication doi: 10.3390/v7112905." (Lines 744-745)
3) Lines 747-749. “Of note, ~50% of gliomas maintain p53 expression therefore limiting the effect of the treatment to half of the tumor cases”. This statement in the context of ONYX-015 has not been proved (see above the comment #2). Remove it.
Authors: The sentence has been removed from the text, as suggested.
4) Lines 753-754 “These modifications make DNX-2401 able to target and replicate within glioma cells lacking Rb expression [324, 325].” The original hypothesis that in normal nonproliferating cells with intact pRb, replication of Ad5 with the delta-24 modification is significantly attenuated (DOI: 10.1038/sj.onc.1203251; DOI: 10.1038/80474) has not not been confirmed by many researchers (please see the introduction and discussion sections in doi: 10.1016/j.omto.2021.12.013)
Authors: Thank you for the comment. We have made the necessary revisions to the text which now read like this: “These modifications were initially believed to enable selective targeting and replication to cancer cells with aberrant Rb pathways. However, other research groups have been unable to replicate these initial findings “
The recommended citation (doi: 10.1016/j.omto.2021.12.013) has been added in the revised version of our manuscript (Lines 755-758)
5) In light of the fact that for a number of viruses the initial hypotheses on the mechanisms of selectivity towards tumor cells were not confirmed later, I would recommend that the authors more carefully study the history of each virus. It is possible that other unproven/contradictory/debatable statements might be identified.
Authors: We have extensively reviewed the mechanisms of selectivity for each virus discussed in the review. We can confirm that with the modifications we have made (refer to responses to points 3 and 4), they are to the best of our knowledge accurate.
6) Standard radiochemotherapy-promoted systemic immunosuppression (iatrogenic) is a significant barrier to immunotherapies. Although different immunosuppressive mechanisms operate in patients with glioblastoma/gliomas at presentation, clinical data provide compelling evidence that the immunological competence of patients may be significantly compromised by standard therapy, exacerbating tumor-related systemic immunosuppression. Standard therapy (radiation, temozolomide, steroids) affects diverse immune cell subsets, with primary immunodeficiency related to long-lasting T cell lymphopenia. Changes in the standard therapy (the need for shifting the treatment paradigm) are required to increase the success of immunotherapies (e.g., discussed in doi: 10.3389/fimmu.2024.1326757). In the section Conclusions and Future Directions, the authors could briefly discuss this issue.
Authors: We thank the reviewer for this valuable comment. In response, we have incorporated two additional sentences into the Conclusions and Future Directions section: “The standard chemoradiotherapy is well-known for inducing immunosuppression and lymphopenia in GBM patients, posing a significant obstacle to GBM immune-based approaches. Essential changes in current standard treatments are crucial for enhancing immunotherapy efficacy (doi: 10.3389/fimmu.2024.1326757)” (Lines 943-946). In addition we provide a new paragraph in which the pros and cons of radiotherapy in combination with immunotherapy are discussed (lines 895-915).
Reviewer 2 Report
Comments and Suggestions for Authors
This review is well-written and offers a large number of organized current trials against GBM. The quality of the figures and tables is also excellent. I recommend that it be published as it is.
Author Response
This review is well-written and offers a large number of organized current trials against GBM. The quality of the figures and tables is also excellent. I recommend that it be published as it is.
Authors: Thank you very much for your very positive feedback. We were happy to know that you liked our review.
Reviewer 3 Report
Comments and Suggestions for Authors
This is an extremely comprehensive review of IO therapy investigations in GBM. I have relatively little to add, except for the comments below:
Minor:
1A. The authors should add a paragraph about radiation therapy in terms of how it interacts with IO. There is one random sentence about this currently. It is the standard of care upfront, and also often used at recurrence. What are the considerations for combining it with IO in either setting? What evidence is there for radiation stimulating IO? In what ways might it counteract IO? Why are some trials combining IO with specific types of radiation therapy?
1. Abstract: current immunotherapies have "so far failed" rather than "failed"
2. 40: suggest adding "primary": "most aggressive primary brain tumor"
3. 46: some trials such as Stupp trial suggest 11% or more may survive 5 years. Suggest adjusting this wording.
4. Line 68: should be "and concomitant AND ADJUVANT chemotherapy..."
5. Line 81: DNA "repair" not "reparation"
6. Line 89: TTF is not that "only" effective therapy, one could also point to a positive trial on carmustine wafers.
7. Line 94: should specify that these median OS and PFS are from the time of recurrence.
8. The authors are pretty definitive in stating that BBB or other factors contribute to treatment failure. But it is hard to say for sure what caused the failures. These claims should be toned down to say for instance "may have caused treatment failure" or "could plausibly contribute to treatment failure" when appropriate.
9. Figure 2 and the text should also mention Herpesviruses such as G207 and results of its investigation in pediatric GBM and adult glioma.
10. Line 926: "(BiTEs) CAR-T cells" should CAR-T cells be removed?
Author Response
This is an extremely comprehensive review of IO therapy investigations in GBM. I have relatively little to add, except for the comments below:
1A. The authors should add a paragraph about radiation therapy in terms of how it interacts with IO. There is one random sentence about this currently. It is the standard of care upfront, and also often used at recurrence. What are the considerations for combining it with IO in either setting? What evidence is there for radiation stimulating IO? In what ways might it counteract IO? Why are some trials combining IO with specific types of radiation therapy?
Authors: We appreciate the reviewer’s comment. As suggested, we added a new paragraph on page 33. This paragraph emphasizes the challenges and opportunities of combining radiotherapy with immunotherapy.” (Lines 895-915). We also cited two new references (369 and 370) which have reviewed the studies on this very active area of research.
- Abstract: current immunotherapies have "so far failed" rather than "failed"
Authors: we have rewritten this sentence, taking into account the comment (Lines 26-27)
- 40: suggest adding "primary": "most aggressive primary brain tumor"
Authors: Corrected, thank you (Line 44).
- 46: some trials such as Stupp trial suggest 11% or more may survive 5 years. Suggest adjusting this wording.
Authors: We have provided the source of our information in a clearer way (Lines 51-53)
- Line 68: should be "and concomitant AND ADJUVANT chemotherapy..."
Authors: we agree (Line 74)
- Line 81: DNA "repair" not "reparation"
Authors: Corrected, thank you (Line 86).
- Line 89: TTF is not that "only" effective therapy, one could also point to a positive trial on carmustine wafers.
Authors: We revised the text from "as the only novel modality" to "a novel modality" (Line 94).
- Line 94: should specify that these median OS and PFS are from the time of recurrence.
Reply: Corrected, thank you (Lines 99-100)
- The authors are pretty definitive in stating that BBB or other factors contribute to treatment failure. But it is hard to say for sure what caused the failures. These claims should be toned down to say for instance "may have caused treatment failure" or "could plausibly contribute to treatment failure" when appropriate.
Reply: We agree with this comment and we have soften down the message throughout the text as suggested. These changes were introduced:
- “The development of effective treatments targeting GBM could plausibly be hampered…” (Line 110)
- “The figure illustrates the distinctive characteristics of GBM (WHO grade 4) that are supposed to hinder the development of effective anti-tumor therapies.” (Line 117)
- “Another GBM key feature that can contribute to treatment failure is the high heterogeneity among (inter-tumoral) and within (intra-tumoral) tumors.” (Line 176)
- Figure 2 and the text should also mention Herpesviruses such as G207 and results of its investigation in pediatric GBM and adult glioma.
Authors: In Figure 2, G207 is already referenced under the virus-based therapy section (green section) within the oncolytic virus category, specifically under the herpes simplex virus family. In Table 4, three concluded clinical trials testing G207 in adult GBM patients are reported. G207 is mentioned in lines 800-802 of the text. As the review focuses solely on adult GBM, clinical trials related to pediatric GBM are not included to maintain this focus and avoid expanding the review further.
- Line 926: "(BiTEs) CAR-T cells" should CAR-T cells be removed?
Authors: Corrected. Thank you (Line 953)